

# Tracer study to estimate the transport of cruise altitude aviation emissions in Northern Hemisphere

Lakshmi Pradeepa Vennam[1,2,3], William Vizuete[2], and Saravanan Arunachalam[1]

[1]Institute for the Environment, University of North Carolina at Chapel Hill, NC, USA.
[2]Gillings School of Global Public Health, University of North Carolina at Chapel Hill, NC, USA.
[3]Now at Ramboll, Novato, CA, USA.

*Correspondence to*: Saravanan Arunachalam (sarav@email.unc.edu)

**Abstract.** Given the increasing role of intercontinental and higher altitude emissions influence on surface air quality, it is important to understand the transport characteristics of these emissions both for policy and mitigation strategies. The

horizontal and vertical transport of directly emitted upper troposphere anthropogenic cruise altitude aircraft emissions (CAAE) has not been well understood due to limited studies. Therefore, in this study we conducted tracer simulations for key source regions in the Northern hemisphere to understand the transport and influence of CAAE on surface air quality. Our results from Northern hemisphere simulations highlight that < 0.6% of CAAE tracer mass fraction occur near the surface even after 90 days of transport time. 30 – 40 % of tracers are found in the upper and mid-troposphere with slightly higher

downward transport occurring during winter than summer season. The tagged source tracer simulations illustrated the source-receptor relationships and showed that ~ 10 – 50% source contributions occur in downwind receptor regions.

## 1 Introduction

The dynamic processes in the atmosphere influence the fate and transport of pollutants in the overall troposphere. In the upper troposphere, due to higher wind speed and westerlies, pollutants can get transported from local regions to regional

and even to transcontinental regions, and vice versa. Several studies (Cooper et al., 2011; Rosa et al., 2012; Lin et al., 2014) investigated the transport of surface emissions to free and upper troposphere, but very limited studies assessed the transport of upper troposphere emissions sources to lower troposphere and surface. One such anthropogenic source that emits pollutants directly into upper troposphere and lower stratosphere (UTLS) region is civil aviation. Aviation is one of the fastest growing modes of transportation with a unique four-dimensional emissions profile. The aviation emissions that occur

between 9 – 12 km are considered cruise altitude emissions (CAAE) and contribute ~ 60 – 75 % (Wilkerson et al., 2010; Olsen et al., 2013) of total aviation emissions in terms of global $NO_X$ and fuel burn. Of all aviation activities, CAAE are predicted to contribute ~ 75% (Yim et al., 2015) of total premature mortalities that are attributed to aviation-related fine particulate matter. Given the ability to transport these emissions over long distances, understanding the fate of these emissions is critical. For example, studies (Liang et al., 2009; Cooper et al., 2011) have highlighted that pollutant levels

increase in highly convective areas and downwind locations due to transport mechanisms and circulation patterns. Further,



several modeling studies (Tarrason et al., 2002; Barrett et al., 2010; Koo et al., 2013; Lee et al., 2013) estimated that the air quality impacts from CAAE emissions (including emissions occurring above ~ 914 m (3000 ft)) at surface are higher when compared to contributions from landing and takeoff (LTO, emissions below ~ 914 m) emissions alone. These studies raise questions regarding our understanding on the role of transport processes on CAAE and their impact on surface air quality.

5      Multiple modeling studies (Gauss et al., 2006; Kohler et al., 2007; Arunachalam et al., 2011; Woody et al., 2011; Vennam et al., 2015; Cameron et al, 2017; Vennam et al, 2017) have investigated the aviation-attributable perturbations due to aircraft emissions from LTO and CAAE at local, regional to global scales and their role in causing human health effects (Stettler et al., 2011; Levy et al., 2012; Morita et al., 2014; Yim et al., 2015). All these studies used the traditional approach to assess the impacts of any individual emission source in atmosphere by calculating differences between 'with emission source (unperturbed)' and 'without emission source (perturbed)' modeling scenarios. With this approach, one cannot isolate the individual role of physical and chemical processes on the overall emissions source impacts in chemistry-transport model. To attribute the sole effect of transport process, an inert tracer modeling approach was implemented previously in few early chemistry-transport studies (Alapaty and Mathur 1998; Allen et al., 1996). In these studies, all the atmospheric processes except for transport processes are turned off. This approach is computationally efficient and presents the opportunity to characterize the transport pathways of an emitted source or source sector. Some recent studies (Wang et al., 2014; Jiao and Flanner 2016) implemented tracer-tagging technique in a global chemistry transport model to quantify source-receptor relationships and transport pathways of black carbon (BC) aerosol. Hence, here in this study we implemented the same tracer approach to study the transport of CAAE at northern hemispheric level and to understand the source-receptor relationships of the CAAE source.

20      Till date, limited studies have looked at the transport of aviation emissions in the UTLS region. An earlier study (Wauben et al., 1997) pointed out that passive transport studies could reproduce the general pattern of aviation $NO_X$ perturbations. Van Velthoven et al., (1997) studied the transport of aviation $NO_X$ passive tracer in an ensemble of models ranging from two-dimensional to three-dimensional chemistry transport and global models. They illustrated that the vertical exchange processes show minor contribution to $NO_X$ concentrations at varying altitudes and models captured these trends. However, the authors clearly identified few limitations where some of the models used in this study lacked parameterization of convective transport, which is crucial for vertical transport and need to be considered in future studies. Hauglustaine et al., (2012) concluded that by treating deep convection in the model, they observed ~10 – 30 % change in surface ozone associated with aircraft emissions. In recent years, global as well as regional-scale model transport schemes were adequately tested, enhanced and uncertainties in the formulations were reduced. Presently, addressing the transport of CAAE with more recent modeling systems with updated transport calculation schemes is ideal. Another study (Whitt et al., 2011) conducted passive tracer simulations by placing CAAE at ~ 11km in the GATOR (Jacobson et al., 2011) global model with $4^o \times 5^o$ horizontal resolution. Their findings concluded that the extra-tropical cruise altitude emissions do not directly affect surface air quality through dynamical vertical mixing processes alone. This study used a coarser model resolution to conduct the passive tracer simulations. On the other hand, multiple studies (Land et al., 2002; Rauscher et al., 2010; Klich et al., 2014;



Gan et al., 2016) have shown better representation of model processes, seasonal trends and local gradients when compared to observational data at a finer vertical and horizontal resolution. Therefore, the transport of CAAE to the surface needs to be further investigated with a finer model resolution (both horizontal and vertical) (Klich and Fuelberg 2014) to improve the understanding of their air quality impacts.

In this study, we characterized the role of dynamic processes in transporting CAAE to the surface by conducting passive tracer simulations. Compared to previous studies our study uses continuous tracer emissions with no decay rate and fresh initialization for each season (further details in methodology). These conditions make this an idealized but worst-case scenario tracer test to quantify the maximum amount of CAAE that can be transported to the surface with continuous emission input and zero loss. Our study also improves upon prior work by using a spatial resolution of $108 \times 108$ km$^2$ which

is $\sim 4 - 5$ times finer than global models ($4^o \times 5^o$). Finally, we also tagged the emissions in three key high aviation activity regions of the world such as North America (NA), Europe (EU) and East Asia (EA) and conducted tagged tracer simulations. To our knowledge, this is the first study to use tagged tracer simulations to illustrate the role of intercontinental transport of civil aviation emissions and to quantify the source region emissions contribution near receptor regions.

## 2 Methodology

### 2.1 Model Inputs and Specifications

       The state-of-the-art Community Multi-Scale Air Quality (CMAQv4.7.1) chemistry-transport model (Byun and Schere 2006) was used over a Northern hemispheric-wide domain at a grid resolution of $108 \times 108$ km$^2$ as shown in Figure 1. For transport schemes, we used the Yamartino (YAMO) (Byun and Schere 2006) scheme for advection process and the Asymmetric Convective Mechanism (ACM2) (Pleim 2007a) scheme for diffusion process. The ACM2 scheme has been

evaluated (Pleim 2007b; Tang et al., 2011) and used in various modeling applications. As mentioned in Pleim et al., (2007a), ACM2 is convective model combined with eddy diffusion scheme that can better represent even sub-grid scale components of turbulent transport in the convective boundary layer.

NASA's Modern-Era Retrospective Reanalysis (MERRA) (Rienecker et al., 2011) meteorology downscaled data was used as inputs to Weather Research and Forecasting model (WRF) (Skamarock et al., 2008) model to generate meteorology data.

Additional description of the WRF configuration and evaluation of the WRF outputs against multiple observation datasets are discussed in Vennam et al., 2017. A finer scale vertical resolution was used in this modelling study to improve the characterization of vertical transport in the model, and avoid potential numerical diffusion. We generated gridded aviation emissions from FAA's Aviation Environmental Design Tool (AEDT) (Roof et al., 2007; Wilkerson et al., 2010) by using a processing tool that spatially and temporally (hourly) allocates the chorded flight segment emissions according to domain's

grid specifications. In this study, we considered only the cruise altitude aviation emissions (CAAE) for year 2006 that fall in the altitude range of $9 - 12$ km.





### 2.2 Tracer simulations

To conduct tracer simulations, we considered emissions of $NO_X$ as our passive tracer since it is one of the highly emitted pollutants at cruise altitudes from aircraft, besides water vapor. The rates of emissions of these tracers were based on actual cruise altitude $NO_X$ emissions estimates from AEDT, to capture the spatial as well as temporal variation of aviation
emissions in the upper layers of the atmosphere.

Tracer simulations were run for three-month periods that coincided with: winter (December – February), spring (March – May), summer (June – August), and autumn (September – November) seasons. For each of the four seasonal simulations, tracers are added throughout the period at cruise altitudes and since there are no removal processes, they will accumulate in the model. Since it typically takes 30 – 90 days for the tropospheric mixing to occur at hemispheric scale (Liang et al., 2009),
this approach enables us to look at the tracer transport processes over a full transport cycle and to assess inter-seasonal variability. These simulations are carried out using cruise altitude emissions in the entire Northern hemisphere for the year 2005.

In addition to the complete Northern hemisphere CAAE tracer simulations, we also conducted another source region-based tracer model scenario, but only using tagged tracers from each of the three regions of largest aviation
emissions: North America (NA: 20N – 60N, 130W – 60W), Europe (EU: 20N – 60N, 10W – 60E), and East Asia (EA: 20N – 60N, 100E –150E). The spatial extents of these regions CAAE are shown in Figure 1. We considered cruise altitude emissions that fall in the spatial bounds of these three sub-regions and named these tracers based on the specific region name (for example: $NO_X\_NA$, $NO_X\_EU$, $NO_X\_EA$). We conducted these simulations for three months modeling period for each season by continuously adding emissions for each of the three sub-regions. The main goal of these tagging tracer runs is to
study the role of intercontinental transport and the impact of each source region on all three receptor regions (i.e., NA, EU and EA).

### 2.3 Analysis metrics

We performed quantitative analysis by calculating mass fraction (MF) in each model layer as shown in equation 1. Equation 1 defines the amount of mass present in each model layer with respect to the total mass available in the model
domain (Column Burden). The output data from the model is usually expressed in concentrations (ppbV), therefore, we converted them into mass basis (molecules/cm$^2$) in each layer (Mass$_{layer}$). We calculated the total mass in the domain by integrating the tracer mass available in all model layers. Overall this mass fraction metric indicates the amount of tracer transported from cruise altitude to different altitudes relative to the total mass burden.

$$\text{Mass Fraction} \left( \text{MF}_{\text{layer}} \right) = \frac{\text{Mass}_{\text{layer}}}{\text{Column Burden}} \times 100 \qquad (1)$$





Source-receptor contributions were estimated by using equation 2 to calculate the fractional tracer contribution of source region emissions at the receptor regions (Wang et al., 2014). In this equation i is the source region (NA, EU, or EA) and j is the receptor region (NA, EU and EA).

$$\text{Contribution}_{i,j} = \frac{\text{Column Burden}_{i,j}}{\sum_{i=1}^{N} \text{Column Burden}_{i,j}} \times 100 \qquad (2)$$

## 3 Results

### 3.1 Tracer surface distributions

The tracer simulations are run over a 90-day period corresponding to the four seasons with a fresh initialization at the beginning of each season. Throughout our calculations, other than where we specifically mentioned each month, we
considered the last month as our representative month for that season. Therefore, all results for a season are based on the last 30 days of the 90-day run. To study the surface tracer distribution, we calculated surface mass fraction percentage using equation 1. When averaged across the domain, the winter season (0.23%) is ~1.6x higher when compared to summer months (0.14%). Winter is followed by autumn (0.21%) and spring (0.18%), both showing ~1.5x and ~1.2x higher than summer (Table 1). The spatial distribution of overall higher tracer mass near the surface in winter (and autumn) is clearly shown in
Figure 2, however the maximum hot spots are seen in summer months near high convection areas.

Figure 2 shows spatial distribution of the surface mass fraction percentage for all months along with maximum domain-wide MF (top corner) that occurred in the spatial extent for each month. Throughout all the seasons (each row in Figure 2), spatially the maximum tracer MF near the surface with respect to the total mass available in the model domain is < 0.6%. The maximum tracer surface MF of 0.56% occurred over the Tibetan Plateau and Middle East region in summer
20 months, whereas in winter season the maximum MF is 0.36%. Though the maximum tracer surface MF is higher in summer the overall average tracer mass near the surface is high during winter. In all seasons, during the first 30 and 60 days, more CAAE tracer mass (> 0.3%) near the surface occurred in the $10 - 40^{\circ}$N latitude bands.

In summer months, the maximum (hot spot) tracer fractions near Middle East and Eastern Mediterranean regions could be due to the tropopause folds (Akritidis et al., 2016) and tracer getting trapped in the subtropical high that descends
near Middle East (Stohl et al., 2002) due to downwelling (downward transport). Additionally, prior studies (Liu et al., 2009) indicated that high concentrations across Northeast Africa and Middle East during summer 2005 in mid-troposphere was associated with anticyclone front transport that descends air from upper troposphere. Due to these episodic transport processes, CAAE tracer from UTLS region was transported to the lower altitudes and surface in these regions. Overall during summer season, the tracer transport occurred mainly at high convection regions and showed maximum tracer
concentrations near these regions. During winter season, horizontal transport dominates due to high westerlies near higher altitudes, therefore the tracers undergo higher intercontinental transport. This higher intercontinental transport also transports



relatively more tracer towards lower altitudes and to the surface layer along the isentropes (Vennam et al., 2017) during winter season. Another interesting pattern during spring and autumn seasons is that higher tracer was seen in the Himalayas and North India region, though these regions have relatively lower aviation emissions. In all likelihood, it is due to higher elevation in this area. This finding explains that some of the CAAE tracer can get transported to remote regions through

vertical transport. In the next section, we discuss the detailed vertical analysis conducted to further understand the transport patterns in higher altitudes.

### 3.2 Tracer vertical distribution

We calculated the relative mass fraction (MF, using equation 1) in each of the modeled vertical layers when compared to the total column burden to understand the transport in the free and upper troposphere regions. Figure 3 shows

the vertical profiles of three months in each season for all model altitudes (each dot indicates model domain vertical layer). Throughout all seasons as expected, the first month has peak MF near $9 - 12$ km as that is where the cruise aviation tracer emissions are emitted. After model simulation with transport process alone for $30 - 90$ days, in second and third months as shown in Figure 3, the peak MF in cruise altitude gradually decreased due to the transport of tracer mass to other model altitudes from the cruise altitudes. Due to this transport pattern, the tracer mass fraction increased in altitudes above and

below the cruise altitudes.

For increased clarity in discussing this vertical analysis, we now refer altitudes bins as regions: $13 - 20$ km (UTLS), $9 - 12$ km (cruise altitude region, CA), $3 - 8$ km (mid-troposphere, MT), $0.05 - 3$ km (lower troposphere, LT) and $0 - 0.05$ km (surface). We summed up the mass fractions (shown in Figure 3) in these regions to quantify the tracer transport. During the first month in each season, 45 (summer) $-$ 50 (winter) % of tracer still remains at cruise altitude region and after 30 days

it decreased to $28 - 40\%$. This change in contribution shows the average transport time of ~1 month near tropopause and cruise altitude region; similar results are also shown in Liang et al., (2009). In winter and autumn seasons, after 3 months, 10 $- 11\%$ of the tracers transported to UTLS, $23 - 29\%$ of tracer remained in CA, 37% transported to MT and $18 - 23\%$ transported to LT. In spring and summer seasons, after 3 months, $14 - 17\%$ of tracer transported to UTLS, $32 - 36\%$ of tracer remained in CA, $33 - 36\%$ appeared in MT and $11 - 15\%$ occur in the LT region. These seasonal differences indicate

that transport of CAAE tracer to mid-troposphere (downward transport) in winter is slightly higher (4%) when compared to summer. In summer, the transport of tracer to UTLS region (upward transport) is slightly higher (7%) when compared to winter and other seasons due to more upward flux. This vertical analysis illustrates that only $0.2 - 0.5\%$ tracer mass reaches the surface even after 90 days of model simulation.

Figure 4 shows the zonal vertical distribution of tracer mixing ratio during the last month of each seasonal

simulation. The tracer data were binned by altitude and by three latitude bands: tropics ($0 - 30$ N), sub-tropics ($30 - 60$ N) and artic ($60 - 90$ N). Consistently in all seasons, the sub-tropics and arctic regions have higher tracer concentrations in CA region (> 9km) and UTLS, since most of the cruise altitude emissions occur in these key source regions (North America, Europe, Middle East and East Asia) and great circle flight paths. As we approach lower altitudes, the tracer concentrations



decreased in the sub-tropics and arctic regions indicating that the CAAE tracer does not directly get transported to the surface in the same latitudes.

In the tropics, tracer concentrations are lower in CA than compared to the sub-tropics and arctic regions due to relatively lower aviation activity. As we reach lower altitudes (3 km) and approach towards the surface, the tracer

concentrations show an increasing trend in the tropics most prominently during winter (autumn) season, indicating that some of the sub-tropics/arctic CAAE tracer was transported to tropics. During summer, relatively higher tracer concentrations in tropics near higher altitudes illustrate the enhanced vertical mixing and upward transport occurring due to warmer weather.

### 3.3 Source-receptor relationships

In this section, we discuss the transport of CAAE from three sub-regions: North America (NA), Europe (EU), and

East Asia (EA). We tagged the CAAE emissions from these regions and looked at each source contribution in the receptor regions. The spatial distribution of CAAE tracer mass fraction percentage (equation 1) near the surface from NA, EU and EA emission tracers are shown in Figures 5 – 7 . The spatial trend clearly indicates that more tracer MF predominantly occurs mainly in the downwind continents (Africa and Asia); however note that the maximum MF at surface is still < 0.6%. For NA tracer (Figure 5), higher tracer MF percentage is in the range of 0.4 – 0.5 % and occurred mainly near India, Indian

Ocean and further towards tropics during winter season. Similar to the NA tracers, the EU tracers shows higher MF percentage near South East Asia, Tibet Plateau and Middle East. The EU tracer (Figure 6) surface maximum MF percentage (0.8%) is ~2x higher than the NA tracer particularly in summer near Tibet Plateau and Middle East. One other interesting observation is that the EU tracer (0.3 – 0.4%) could get transported to western NA particularly in winter and autumn seasons through Trans-Pacific synoptic transport due to strong westerly transport in the cruise altitude region. With EA tracer (Figure

7), the maximum MF percentage of 0.4% occurred in the Pacific region during spring season. This strong westward transport in spring from EA was already observed in previous studies (Lin et al., 2012) and we are seeing a similar transport pattern with the EA CAAE tracer simulation.

Figure 8 shows the contribution of the three source regions near the receptor regions computed using Equation 2. In NA receptor region, 46 – 50% of NA contribution is due to NA tracer emissions and the remaining 28 – 34% and 19 - 21% is

due to EU and EA tracer emissions in all seasons except in summer. In summer, 69% is due to NA emissions with the remaining 21% and 11% contributions incurred from EU and EA tracer emissions. This high contribution in summer is due to the transport of NA emissions to the surface near high convection regions like western NA. Whereas in the EU receptor region, 43 – 48 % contribution is due to NA source followed by 36 - 41% and 15% contributions from EU and EA sources for all seasons except summer. During summer, both NA and EU tracers show equal contribution of 45% and EA shows 9%

in the EU receptor region. This highlights that in EU, the influence from NA is prominent in all seasons due to strong westerly transport. In EA receptor region, 40 – 45% contribution occurred from NA and 33 – 37% contribution came from EU source followed by EA source with 22 – 24% in all seasons except summer. In summer, the NA source contribution decreased to 29% whereas EU and EA contribution increased to 40% and 31%.



Throughout all three receptor regions, their own source contribution increased in summer season indicating the influence of vertical transport due to relatively high convective mixing when compared to all other seasons. During winter and spring seasons, the downwind source region contributions are higher indicating the influence of westerlies and horizontal transport at higher altitudes.

## 5  4 Conclusions

In this study, we implemented a passive tracer approach to understand the role of physical processes in transporting cruise altitude aviation emissions (CAAE) in the atmosphere. Overall the model predictions indicated < 0.6% of CAAE in the total column was transported to the surface in northern hemisphere for all seasons, indicating that most of the tracer still exists in the MT and UT. Therefore, more than direct transport to surface, most of the CAAE tracer is transported to MT region and can be subjected to the chemical processes in these regions. CAAE tracers tend to concentrate in sub-tropics and arctic region at cruise altitudes. As we approach the surface, however, tracer concentrations began to increase in the tropic regions. This is the result of the model transporting CAAE tracers from the sub-tropics higher altitudes towards lower altitudes in the tropics. We found that tracers in upper altitudes are mainly driven by horizontal transport followed by vertical transport at high downward flux convection regions. Winter season shows higher proportion of tracer mass near the surface than summer season, whereas summer season showed maximum tracer near high convection regions.

From our source-receptor analysis we found that both NA and EU regions are primarily impacted by emissions from their own regions. Overall, we see that NA source emissions can significantly affect EU and EA regions in all seasons and both NA as well as EU source emissions can affect EA region. Our intercontinental tracer study showed evidence that NA and EU cruise emissions mainly show higher impacts near high terrain regions like Tibet Plateau and can also impact places with relatively lower aviation emissions regions such as North Africa, India and South East Asia due to transport. This partly explains some of the aviation-attributable high mortality estimated by Barrett et al., (2010) in Asia. A future extension of this work is to go beyond the physical processes, and study the role of chemical processes in UTLS region by conducting sensitivity analyses with different chemical mechanisms. Note that the range of mass fraction provided in this study cannot be directly translated to concentrations or surface air quality impacts, as we did not consider the atmospheric chemical processes in our modeling study. However, the key purpose of this study is to highlight the extent of CAAE transport to the surface and lower altitudes when the worst-case scenario (no chemistry, no decay or loss) is considered. The tracer mass contributions presented in this study is entirely driven by the meteorology considered in the modeling set up, and these CAAE tracer contributions are subjected to change with meteorological patterns from other years, and other model or model configurations (due to change in transport schemes).



**Acknowledgements**

This work was funded by the Federal Aviation Administration through grants under the Partnership for AiR Transportation Noise & Emissions Reduction (PARTNER) (http://partner.mit.edu) and Aviation Sustainability Center (ASCENT) (http://ascent.aero ) to the University of North Carolina at Chapel Hill
under 13-C-AJFE-UNC. PARTNER and ASCENT are FAA/NASA/Transport Canada/US DOD/EPA-sponsored Centers of Excellence. The aircraft emissions inventories used for this work were provided by the U.S. Department of Transportation's Volpe Center. Any opinions, finding, and conclusions or recommendations expressed in this work are those of the author(s) and do not necessarily reflect the views of FAA or Volpe. We would also like to acknowledge Bok Haeng Baek from the UNC Institute
for the Environment for developing the AEDTProc tool, and Mohammad Omary for initial assistance with processing the AEDT emissions for CMAQ.

**Data Availability**

The readers are requested to contact the corresponding author for the data that supports the analysis and conclusions of this work.

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



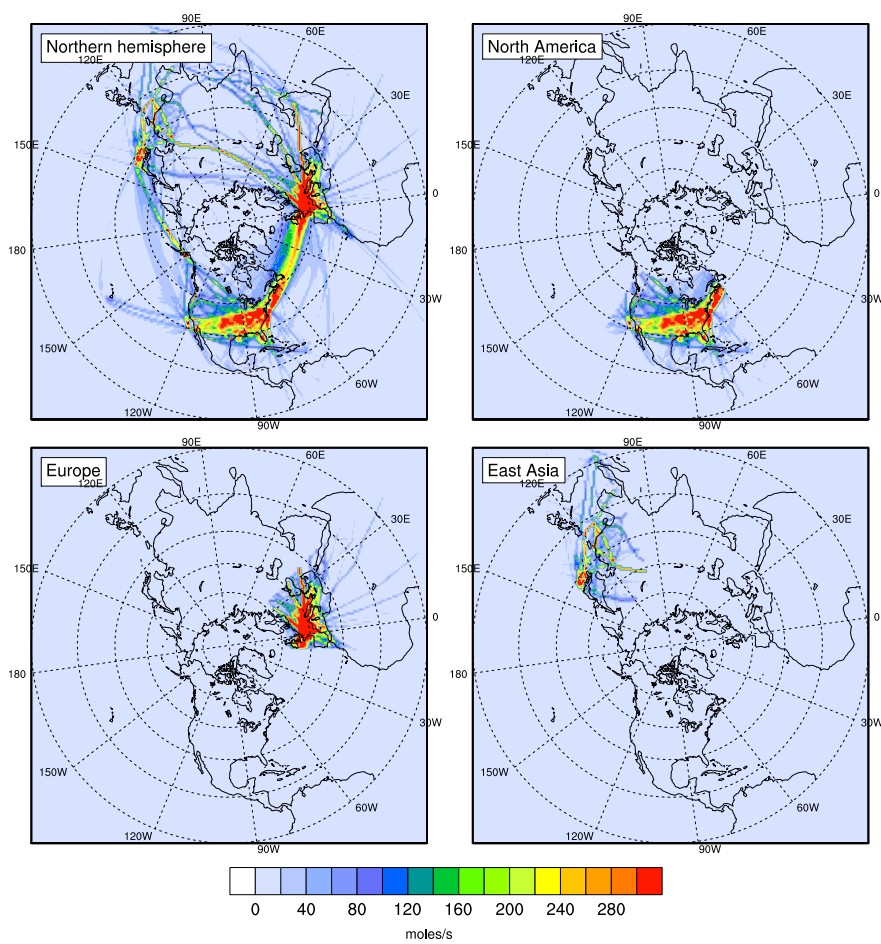

**Figure 1: Hemispheric modeling domain with cruise altitude emissions distributions for complete Northern hemisphere (top, left). Also shown are the tagged cruise aviation emission scenarios for North America (NA) (top, right), Europe (EU) (bottom, left) and East Asia (EA) (bottom, right).**




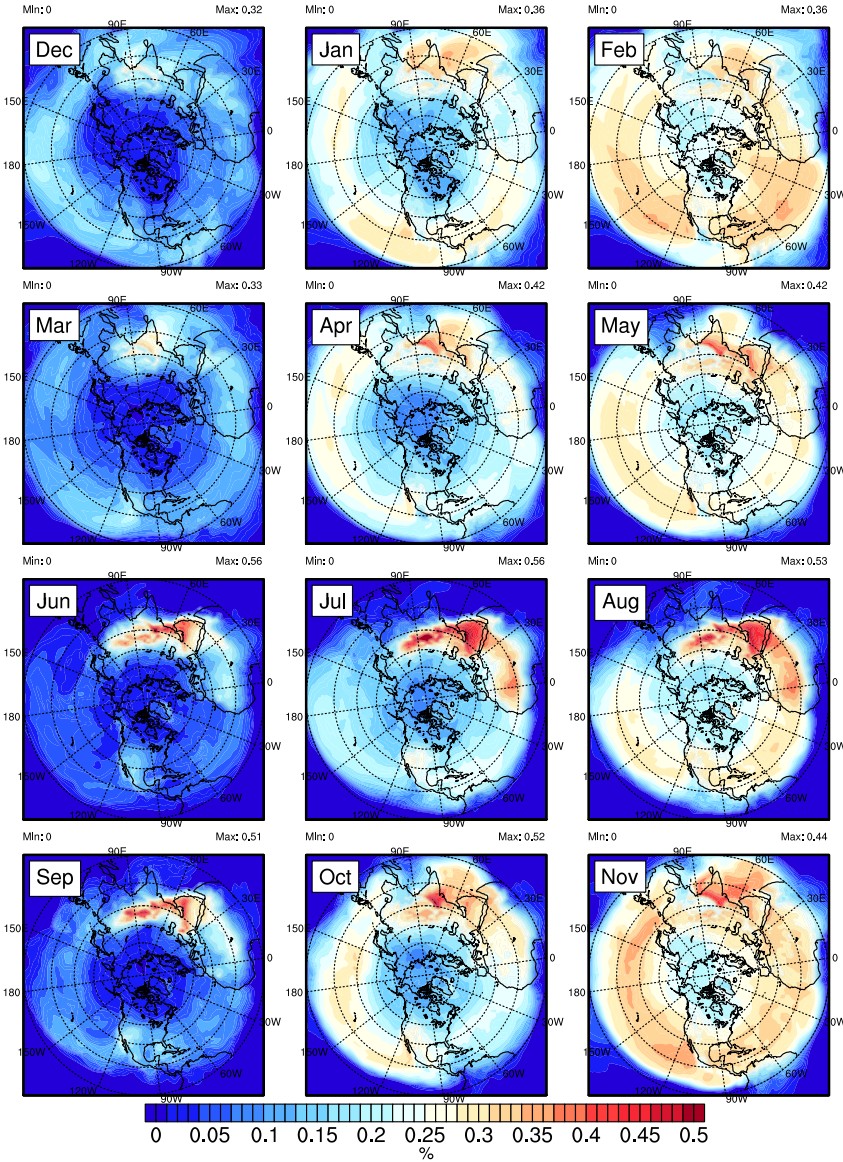

**Figure 2:** Tracer surface mass fraction percentage with respect to the total mass available during each month. Each row is a single simulation where tracers were reset to zero at the beginning of the first month of each row, and represents one season.





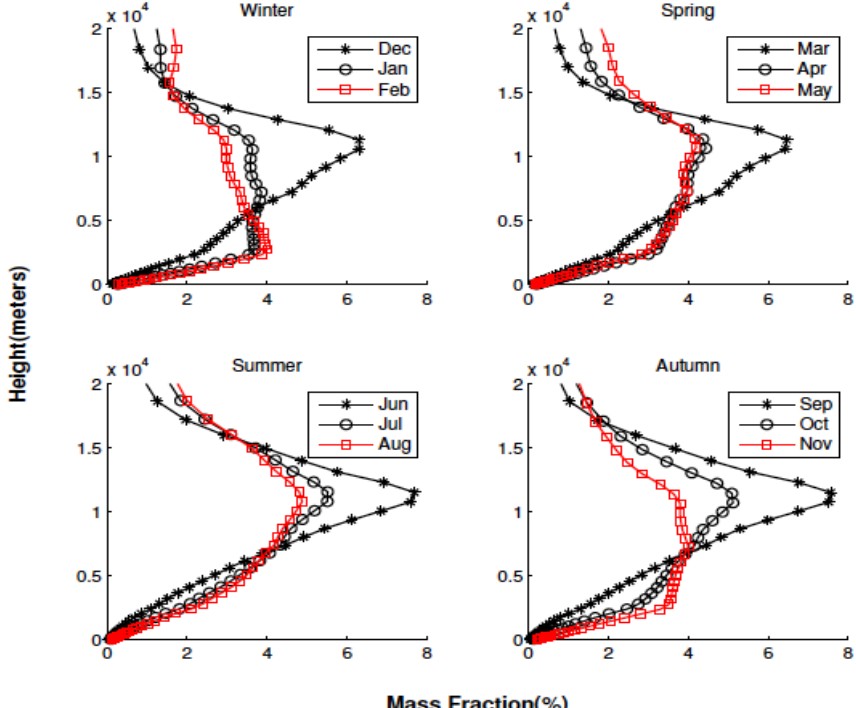

**Figure 3: The overall model domain vertical profile of tracer mass fraction (%) at different altitudes (points on the line) in the model for each month in a season.**





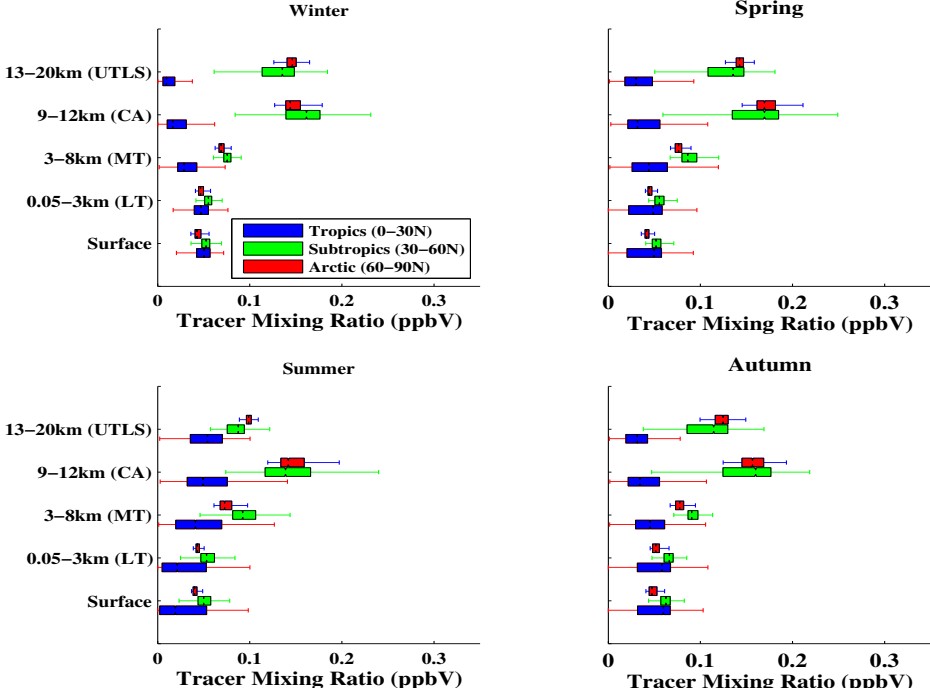

**Figure 4: Monthly averaged tracer mixing ratios for the last month in each season. Each box plot represents the tracer mixing**
5 **ratios of all grid cells that fall in latitude bins of 0 – 30N (blue), 30 – 60N (green), and 60 – 90N(red). The latitude bins based**
**mixing ratios are further binned vertically by altitudes.**



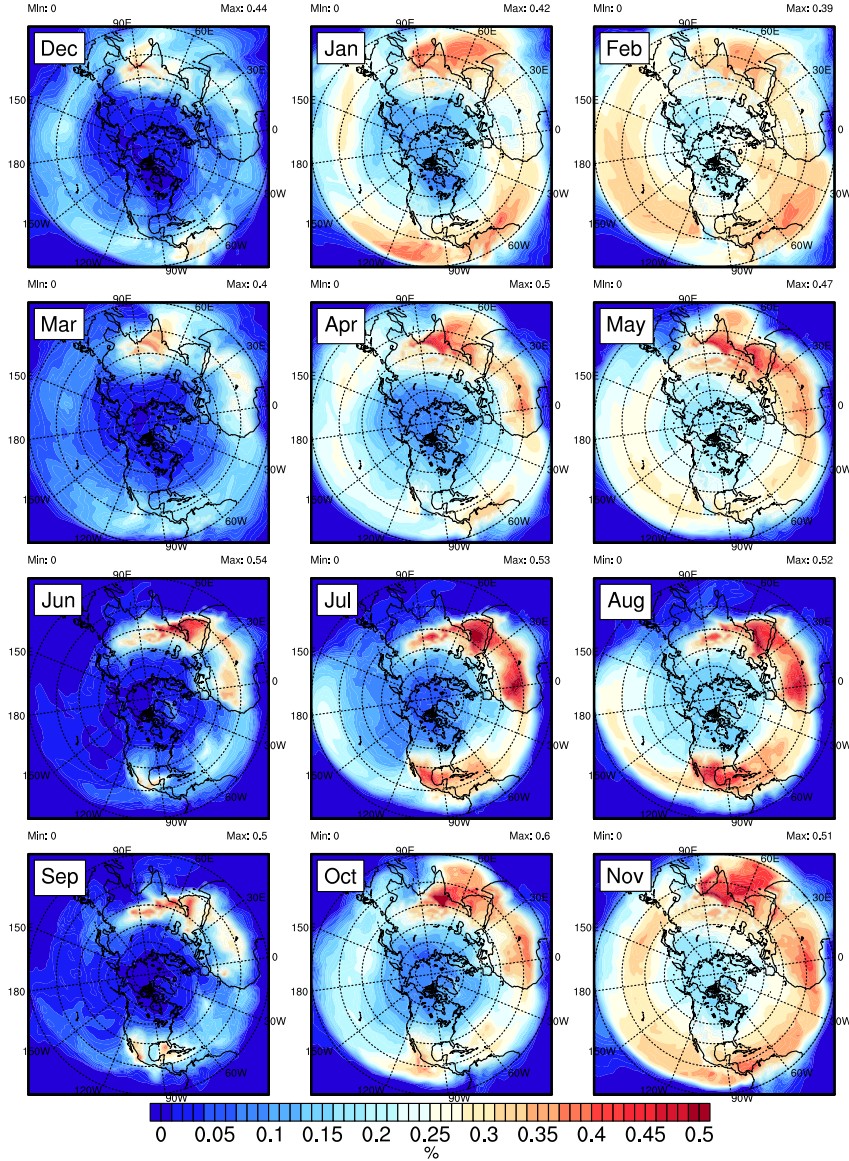

**Figure 5: North America tracer surface mass fraction percentage with respect to the total mass available in the model domain for each month. Each row is a single simulation where tracers were reset to zero at the beginning of the first month of each row, and represents one season.**





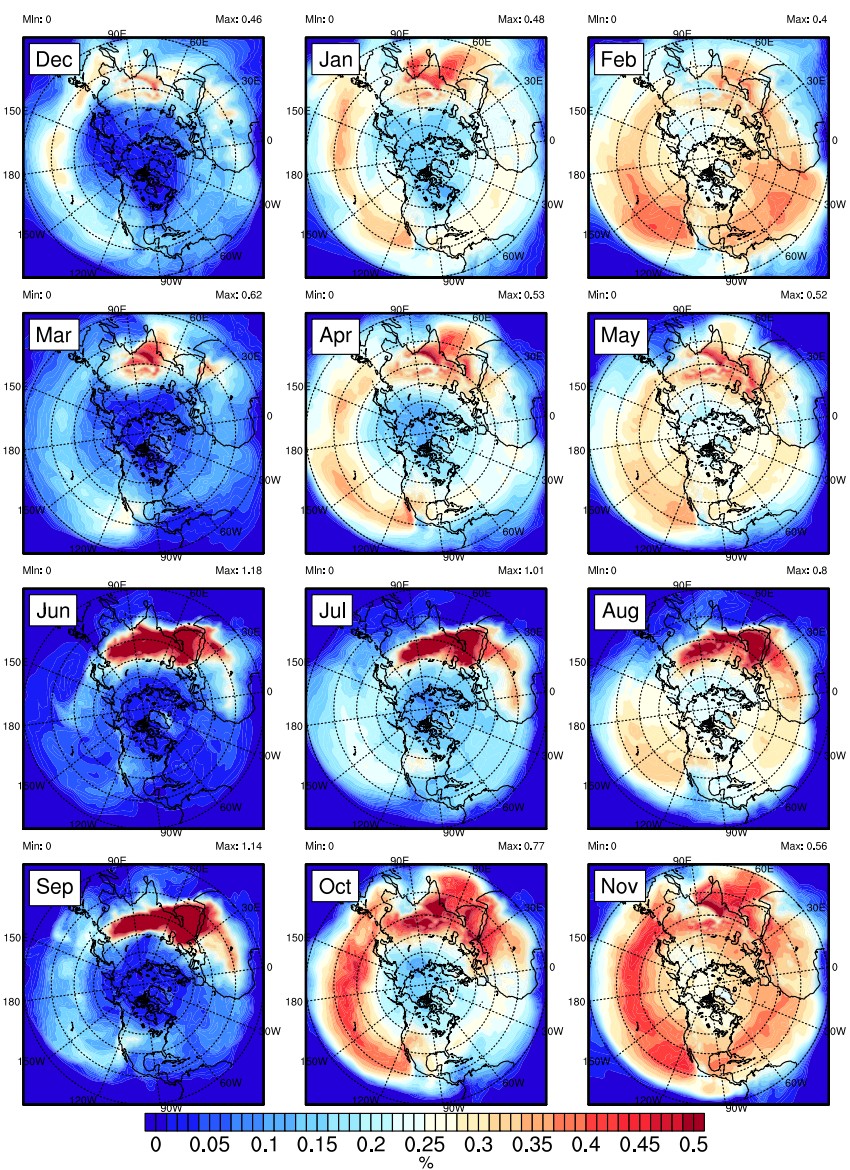

Figure 6: Same as Figure 5 but for Europe Tracer.





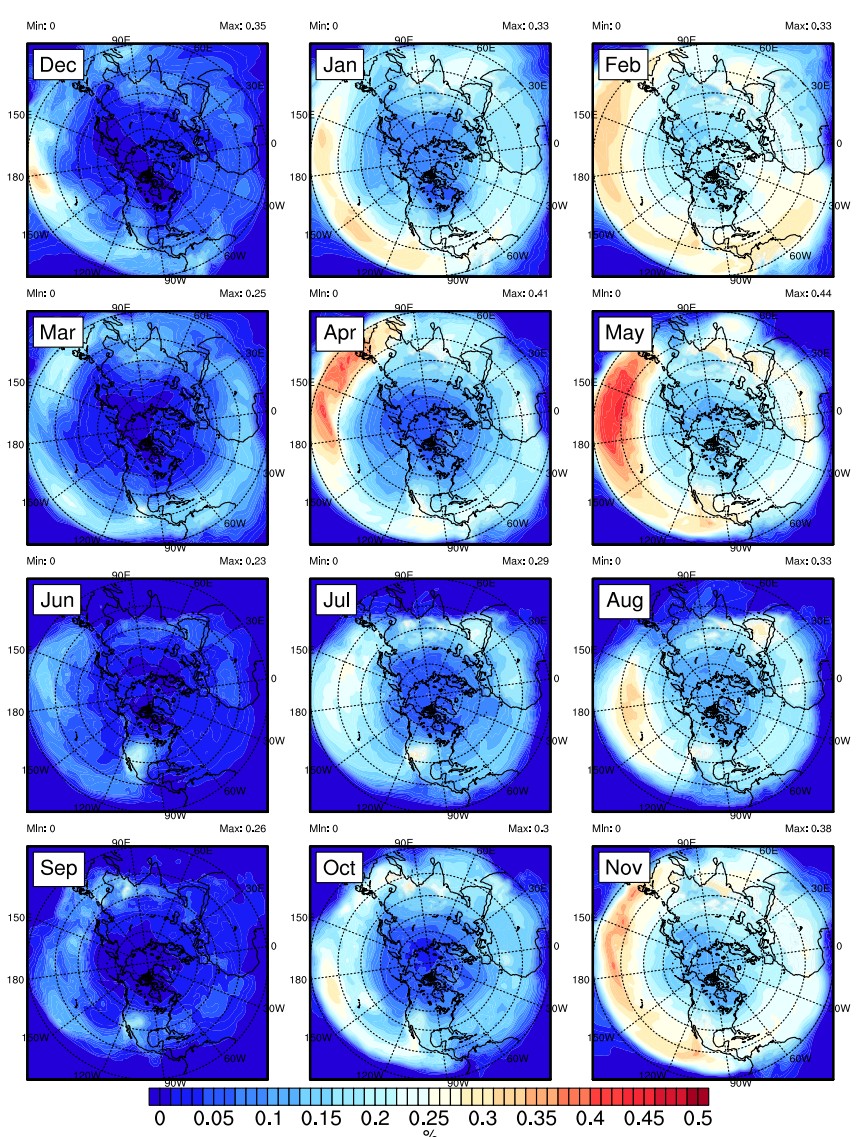

**Figure 7: Same as Figure 5 but for East Asia Tracer.**



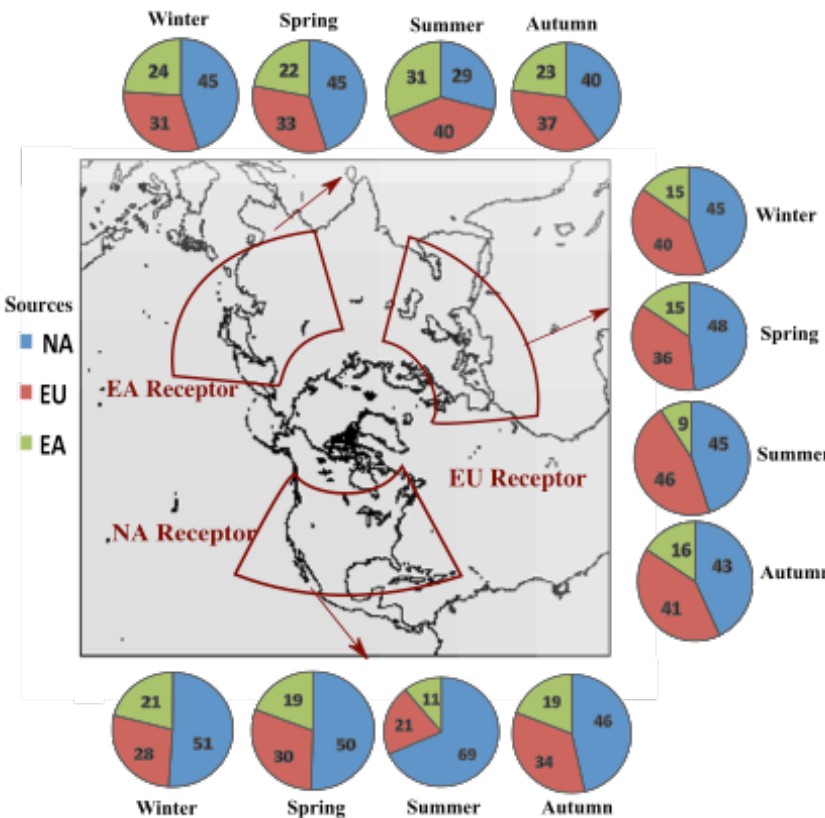

**Figure 8: Seasonal tracer source-receptor contribution metric (Equation 2) (%) for North America (NA), Europe (EU) and East Asia (EA) sources.**



**Table 1: Tracer source-receptor contribution metric (Equation 2) (%) for four seasons of North America (NA), Europe (EU) and East Asia (EA) sources.**

| Season | Surface Mass Fraction (%) |
|--------|---------------------------|
| Winter | 0.23 |
| Spring | 0.18 |
| Summer | 0.14 |
| Autumn | 0.21 |