# Peer review of "Tracer study to estimate the transport of cruise altitude aviation emissions in Northern Hemisphere"

_Atmospheric Chemistry and Physics, 2018_

## Short Comment (SC1) · 10 Sep 2018

The authors may be interested in an earlier tracer study of aircraft emissions that seems relevant for the current study:

Forster, C., A. Stohl, P. James, and V. Thouret (2003): The residence times of aircraft emissions in the stratosphere using a mean emissions inventory and emissions along actual flight tracks. STACCATO special section of J. Geophys. Res. 108, 8524, doi:10.1029/2002JD002515.

[Figure]

2018.

---

## Referee Comment (RC1) · Anonymous Referee #1 · 12 Sep 2018

**General comment**

The study by Vennam et al aims at understanding the transport of species emitted by aviation with special focus on their impact on near surface values. This is an important topic and the idea to tackle the problem with a simplified numerical simulation is appealing. The paper is well written and in principal suitable for ACP. While having said this, I unfortunately have severe concerns with respect to the applied model, simulation set-up, presented results and the interpretation of the results. While the issue with the suitability of the model for this specific application might only be a lack of information (I tried to find respective results in other publication, but couldn't find them), the sim-

ulation set-up is, as far as I can judge, not supporting what the authors wish to show. More detailed information is given below. I recommend that the authors revise their manuscript with a more adequate simulation set-up and results presentation, which basically leads to a new submission.

Major Comments:

A) Model. Currently I am not able to judge whether the model is able to adequately model the dynamics of the tropopause layer. From Figure 3 I guess that the model vertical resolution is around 1 km and the model top is at 20 km. Please give a reasoning why CMAQ is capable to correctly treat the transport of species emitted at around 10 km. There is some indication that quite some counter-gradient transport might happen, since the model transports quite some amount of the tracer to altitudes of around 20 km, exceeding the surface values (Figure 3). How do you explain this transport pathway to such high altitudes in such short time?

B) While understanding the experimental set-up, I do not understand how to correctly interpret the simulation results. The quantity "Mass Fraction(surface layer)" is to zero at the beginning of the simulation. During the simulation the values increase constantly (as explained by the authors). The concentration increases everywhere in the model domain without limits and the ratio between the concentrations in the surface model layer to the column will converge to the model layer air mass to the column air mass and hence independent from the emission location. Latter because the difference between concentrations of model grid points concentration is getting small compared to the absolute steadily increasing concentration. Hence it looks arbitrary to me to take out any specific point in time. I think there is a principle problem in the interpretation of the results for this simulation set-up, which can only be resolved by a change in the simulation set-up. One possibility is to define a sink at the surface (deposition, ...) in a meaningful way. The simulation will converge to a quasi-steady state. This has also the advantage of having the possibility to check whether the results are in steady-state after three months. Referring to Grewe et al 2014 Figure 9, the water vapour

temporal evolution for cruise emissions show larger time scales, which might question the assumption of achieving a steady-state after 3 months (or actually 2 months, see below).

C) Quantities are biased by the model resolution. Results are not given independently from the model resolution (see below), which inhibits a proper interpretation, even if the comment B wouldn't apply.

Specific comments Abstract

p1 / l11 Please specify a bit more what kind of tracer you are referring to: inert gas-phase, inert particles with sedimentation, ..., what loss processes?

p1 / l12 Please explain in more detail what "tracer mass fraction means" in this context. Everything which is emitted in the atmosphere will eventually be deposited at the ground. From this perspective 100% would be expected. Near the surface is crucial. The smaller the volume the lower the percentage?

p1/l14 why "even"? It seems that the authors have expected something else. Please clarify this. p1/l16 Unclear. If something is emitted at 12 km it will always be deposited downwind. There is no direct instantaneous downward transport.

Introduction

p1/l20-22 There are indeed a couple of passive tracer studies, which were not included here, but a comparison might have been of interest. They are often not directly referring to the surface as a receptor region, but Figures are often including this information.

* Velthoven et al Atmospheic Environment 1997, is referred to below.

* Köhler, I., Sausen, R., Reinberger, R., Contributions of aircraft emissions to the atmospheric NOx content, Atmospheric Environment, Volume 31, Issue 12, 1997, Pages 1801-1818,

* Danilin, et al. Aviation fuel tracer simulation: model intercomparison and implications,

Geophys. Res. Lett., 25 (1998), pp. 3947-3950

* Schoeberl, M.R., Morris, G.A., 2000. A Lagrangian simulation of supersonic and subsonic aircraft exhaust emissions. Journal of Geophysical Research 105, 11,833–11,839

* Rogers, H. L., H. Teyssedere, G. Pitari, V. Grewe, P. van Velthoven, J. Sundet, Model intercomparison of the transport of aircraft-like emissions from sub- and supersonic aircraft, Meteorol. Z., 3, 151-159, 2002.

* Grewe, V., Reithmeier, C. and D.T. Shindell, Dynamic-chemical coupling of the upper troposphere and lower stratosphere region, Chemosphere: Global Change Science, 47, 851-861, 2002.

Some of this is referred to later in the text, but should be clarified already here, since the impression is given that those studies do not exist.

p2 / l5-19 The text might suggest that aircraft emission tagging approaches were not used previously. However, there are studies 20 years back (Brasseur et al 1998) or recently Grewe et al 2017, who use such approaches. Please clarify the text.

* Brasseur et al, European scientific assessment of the atmospheric effects of aircraft emissions, Atmospheric Environment, 32, 1998, 2329-2418 (Figure 35)

* Grewe, V., Tsati, E., Mertens, M., Frömming, C., and Jöckel, P., Contribution of emissions to concentrations: The TAGGING 1.0 submodel based on the Modular Earth Submodel System (MESSy 2.52), Geosci. Model Dev. 10, 2615-2633, doi:10.5194/gmd-2016-298, 2017.

p3/l5 I suggest to use "atmospheric transport (resolved, parameterised and unresolved)" instead of "dynamic", since the role of turbulence, diabatic heating, etc. is not investigated.

p3 l7-9 I do not understand the importance and significance of this approach. There

are several effects mixed. Continuous emission lead to continuous increase of the concentrations. At any time, the surface concentration is a snap shot and mixes emission at time T0, where a lot of the emitted species may have reached the surface with emissions at time T0+90days, where no contribution to the surface concentration is expected. Further, since the tracer is not deposited it remains in the atmosphere and artificially increases the concentration, which disagrees with the worst case assumption.

Methodology

Section 2.1: Nothing is said about the vertical extend of the domain, number of layers and especially the resolution at tropopause levels. This is an extremely important point. Large-scale transport as well as diffusivity of the transport scheme might give largely different results for too low vertical resolution or a too low model top. The large concentrations at 20 km might indicate such problems.

Section 2.2 p4 l 14

why "only"? Is there a disadvantage in this approach?

Equation (1):

I strongly advise the authors to convert this mass fraction from an extensive to an intensive quantity. In the current version "Mass Fraction (MF-layer)" is dependent on the model resolution, which is not given (see above). For example a thick layer at 500 hPa will give a large number, not because there is a lot of tracer mass in terms of concentration, but solely because the model layer is thick. One possibility is to further divide by the layer thickness to obtain %/km as unit. The consequence is that the results are resolution independent, better to be interpreted and can be compared to other model results. In Figure 3, e.g., the total amount can be obtained by summing up the values. When MF is divided by the layer thickness, the total amount can be obtained by integration.

[Figure]

Results

p5l11 I think, by taking the last 30 days out of a 90 days simulation period, you implicitly assume a spin-up of 60 days and 30 days simulation, right? Hence, actually you assume a much smaller mixing time than suggested in Section 2. I suggest being a little bit more specific about the lifetimes used. For example if you assume that 95% of the air is mixed after 60 days, this will result in an e-folding mixing lifetime of 20 days, which is certainly too low. A 60 day e-folding mixing lifetime will result in a 95% mixing after 180 days, which would require a much longer simulation time.

Figure 1: Units are actually moles/s/gridbox. Please provide Figures, which are not dependent of the chosen resolution, e.g. moles/s/m2 or moles/s/m3.

p5 l23: How well do you simulate tropopause folds and the associated strat-trop exchange?

p5 l23ff: The discussion is very speculative. I think you should be able to support your arguments with your simulation results. E.g. showing the tropopause fold etc.

---

## Referee Comment (RC2) · Anonymous Referee #2 · 13 Jan 2019

The paper discusses the dispersion of emissions of aviation from cruise altitudes in the Northern Hemisphere atmosphere. The paper considers the emissions as given in a data set as provided by FAA and Volpe for the ACCRI project [Wilkerson et al., 2010; Brasseur et al., 2016]. The emissions are treated as passive tracers, without any removal process in the atmosphere. The emission are followed over several seasons (3 months periods), starting from zero initial concentrations. The model considers tracer transport by advection with the resolved wind field of a hemispheric global model and by diffusion from a convective mechanism. I do not know this model and information is given in this paper about this model only in terms of a few references and some resolution information.

[Figure]

So the paper studies how a passive tracer emitted from a more or less continuous source near the tropopause accumulates in and gets distributed over the atmosphere within 3 months periods for various seasonal meteorological conditions.

The paper studies the mass fraction of tracers in model layers and in various source and receptor domains including a surface layer (of unknown vertical thickness).

The paper aims to investigate physical processes in transporting cruise altitude emissions in the atmosphere. However, I cannot learn anything about physical processes except that they vary with season and altitude, and that convection may be important in summer. That is not new.

The paper claims to be the first in using a "tagged tracer simulation" to quantify source-receptor relationships. Tagging is needed to follow the fate of tracers in a nonlinear system [Grewe et al., 2010]. In this study, the tracer transport is linear in the concentration values. A doubling of the sources causes a doubling of the concentrations. In this case, emissions from various sources can be treated independently of each other and tagging is trivial. Similar studies of the dispersion of NOx as a passive tracer from various sources, with linear chemistry, have been presented, e.g. by Ehhalt et al. [Ehhalt et al., 1992] and Köhler et al. [Köhler et al., 1997], long ago.

So, this is an academic study. That would be acceptable if done well. However, I also have technical problems:

What is the vertical resolution. How thick is the surface layer? What are the time step sizes? Which process is simulated by asymmetric diffusion?

Page 4, line 3: why do you mention water vapor. Why not CO2?

More general, why do you talk about NOx emissions when you simulate the emissions as a passive tracer? NOx has a lifetime of typically 5 days in the free troposphere, and often much shorter near the surface. Thereafter, most NOx is converted to HNO3 and other species after a few days. CO2 would be closer to the passive tracer concept.

I am sceptical about the conservation properties of the model in this study. The paper talks about the amount of a species measured in moles. But I would expect that one should discuss a conservative concentration measure like the molar mixing ratio of the tracer (number of moles of the tracer per mole of air).

Fig. 1 presents emissions in units of moles/s. This is a species abundance source rate. In order to assess this, one needs to know the respective air volume in which the emissions occur.

When computing mean values, do the authors weigh the results with the volumes or do they add concentrations from small grid cells (near the poles and near the surface) with the same weight as sources from large grid cells (in the tropics and in the upper atmosphere)? That is not clear presently.

Fig. 3, winter, shows a maximum of mass fraction forming in the lower troposphere, i.e., in a region without sources. How can that happen? Yes it can happen temporarily when advection dominates relative to diffusion processes. When averaged over longer period, it should not happen. However that is not discussed. I have the impression that the model violates conservation laws.

The paper is good in citing many related studies. In fact, I was not aware on many of them. But it appears somewhat random in the selection of references (those of major and minor relevance for this paper). There are many other important studies which dealt with tracer or aviation emission transport in the global atmosphere earlier or more complete. Examples are as follows: Ehhalt et al. [1992]; Danilin et al. [1998] , Forster et al. [2003], Koehler et al. [1997], Brasseur et al. [1996], Brasseur et al. [1998], Gauss et al. [2006].

In summary, the paper in its present form does not satisfy the quality criteria of ACP.

I just looked at the paper Veenam et al. (JGR, 2017), cited in this paper, which just appeared. It seems that this is far more advanced. It includes the chemical processes

that are mentioned in the outlook of the ACPD paper. So, I am not convinced that the present paper is still needed.

References

Brasseur, G. P., J.-F. Müller, and C. Granier (1996), Atmospheric impact of NOx emissions by subsonic aircraft: A three-dimensional model study, J. Geophys. Res., 101, 1423-1428, doi: 10.1029/95JD02363.

Brasseur, G. P., R. A. Cox, D. Hauglustaine, I. Isaksen, J. Lelieveld, D. H. Lister, R. Sausen, U. Schumann, A. Wahner, and P. Wiesen (1998), European scientific assessment of the atmospheric effects of aircraft emissions, Atmos. Env., 32, 2329 - 2418. Brasseur, G. P., et al. (2016), Impact of aviation on climate: FAA's Aviation Climate Change Research Initiative (ACCRI) Phase II, Bull. Amer. Meteorol. Soc., 97, 561-583, doi: 10.1175/BAMS-D-13-00089.1.

Danilin, M. Y., et al. (1998), Aviation Fuel Tracer Simulation: Model Intercomparison and Implications, Geophys. Res. Lett., 25, 3947 - 3950.

Ehhalt, D. H., F. Rohrer, and A. Wahner (1992), Sources and distribution of NOx in the upper troposphere at northern mid-latitudes, J. Geophys. Res., 97, 3725 - 3738, doi: 10.1029/91JD03081.

Forster, C., A. Stohl, P. James, and V. Thouret (2003), The residence times of aircraft emissions in the stratosphere using a mean emission inventory and emissions along actual flight tracks, J. Geophys. Res., 108, 8524, doi: 10.1029/2002JD002515.

Gauss, M., I. S. A. Isaksen, D. S. Lee, and O. A. Søvde (2006), Impact of aircraft NOx emissions on the atmosphere – tradeoffs to reduce the impact, Atmos. Chem. Phys., 6, 1529–1548.

Grewe, V., T. Eleni, and P. Hoor (2010), On the attribution of contributions of atmospheric trace gases to emissions in atmospheric model applications, Geosci. Model Dev., 3, 487-499, doi: 10.5194/gmd-3-487-2010.

Koehler, I., R. Sausen, and R. Reinberger (1997), Contributions of aircraft emissions to the atmospheric NOx content, Atmos. Env., 31, 1801-1818.

Wilkerson, J. T., M. Z. Jacobson, A. Malwitz, S. Balasubramanian, R. Wayson, G. Fleming, A. D. Naiman, and S. K. Lele (2010), Analysis of emission data from global commercial aviation: 2004 and 2006, Atmos. Chem. Phys., 10, 6391-6408, doi: 10.5194/acp-10-6391-2010.

---

## Author Comment (AC1) · 21 Apr 2019

Comment: The authors may be interested in an earlier tracer study of aircraft emissions
that seems relevant for the current study:

Forster, C., A. Stohl, P. James, and V. Thouret (2003): The residence times of air- craft
emissions in the stratosphere using a mean emissions inventory and emissions along
actual flight tracks. STACCATO special section of J. Geophys. Res. 108, 8524,
doi:10.1029/2002JD002515.

Response: We thank the reviewer for pointing us to this tracer study of aircraft emissions.
Forster et al., studied the residence times of aircraft emissions in the stratosphere using a
mean emissions inventory in the North Atlantic Flight Corridor ( NAFC). The authors
comprehensively described the residence time using the $NO_x$ passive tracer age and also
showed the difference between using mean emissions inventory and inventory along
actual flight tracks. While Forster et al., studied a very important topic it is slightly
different from our study in some aspects:
1) Our study focused on the tracer mass that reached the surface from the cruise
   altitudes due to transport
2) We used highly resolved aircraft emissions and looked at the entire northern
   hemisphere rather than just the Atlantic corridor.
3) We discussed the overall vertical profile of the tracer in the 3-month model
   simulation time and how it is varying seasonally
4) We also showed how the tracer contributions are varying at three sub-regions
   (Tropics, sub-tropics and arctic)
5) Finally we discussed the CAAE tracer source-receptor relationships at North
   America (NA), Europe (EU) and East Asia (EA) regions near the surface.

Based on our reading we agree that it is a relevant reference for our present study, so
included it in the revised manuscript and added the following lines "*Another study
(Forster et al., 2003) investigated the residence times of the North Atlantic Flight
Corridor (NAFC) using a Lagrangian dispersion model and showed that the
stratospheric emissions are transported in northeasterly directions with maximum flux
occurring near Europe and North Africa, however this study focused only on the NAFC
and used a mean emission inventory*".

---

## Author Comment (AC2) · 21 Apr 2019

The authors are thankful to the reviewer for a thorough review and for raising several interesting and valid points that provided us an opportunity to clarify several aspects of this manuscript and improve it overall. Below are our responses to the reviewer comments and revisions to the manuscript when applicable.

*Comment:* Currently I am not able to judge whether the model is able to adequately model the dynamics of the tropopause layer. From Figure 3 I guess that the model vertical resolution is around 1 km and the model top is at 20 km. Please give a reasoning why CMAQ is capable to correctly treat the transport of species emitted at around 10 km. There is some indication that quite some counter-gradient transport might happen, since the model transports quite some amount of the tracer to altitudes of around 20 km, exceeding the surface values (Figure 3). How do you explain this transport pathway to such high altitudes in such short time?

*Response:* The CMAQ model has been used for numerous regional and global applications and is capable of fully capturing horizontal and vertical transport near tropopause and upper troposphere. In this study the model layer top is ~50 mbar and the vertical structure is similar to recent hemispheric CMAQ (H-CMAQ) studies (Mathur et al., 2017; Hogrefe et al., 2018), and now included as a table in Supplementary information. We used the same WRF meteorological data as were used in these prior studies with no layer collapsing which maximizes the consistency between H-CMAQ and WRF, and further reduces the vertical diffusion of the inter-continental transport. The UTLS dynamics are highly dependent on the reanalysis data used in the nudging and further details of the WRF configuration are discussed in Xing et al., 2015. Additionally in this application, higher vertical resolution (< 1km ) was also used above the boundary layer (i.e., free and upper troposphere) than traditional CMAQ regional-scale applications to better resolve the tropopause dynamics. Mathur et al., 2017 briefly discussed the motivation behind 50 mbar model top (44 model layer structure) and highlighted that this layer structure is less diffusive in entraining the free-troposphere tracers to boundary layer. Recent hemispheric CMAQ studies extensively evaluated the free-troposphere and upper troposphere vertical profiles of various pollutant concentrations with in-situ aircraft measurements (Mathur et al., 2017; Vennam et al., 2017) and ozonesonde data (Xing et al., 2016; Hogrefe et al., 2018). These studies have indicated that H-CMAQ shows good vertical representation in the UTLS region when compared with observations, and vertical profiles look comparable to some other global models that consider the fully coupled stratosphere-troposphere dynamics. Overall this discussion reinforces our justification that H-CMAQ is capable of capturing the tropopause and upper troposphere

transport which is the central focus of this study. However, we acknowledge and mentioned these lines in the revised manuscript *"we acknowledge that there is no detailed assessment of the UTLS dynamics with this configuration and recommend that this be investigated further in future H-CMAQ studies."*

We attribute the upward transport partly to the strong winds aloft that can horizontally transport and can eventually vertically advect some of the tracer mass to higher model layers. Also note that in Figure 3 we are averaging all horizontal grid cells concentrations at each altitude to calculate the mass fraction at each layer. Thus, in few grid cells if there is a upward draft it can transport the tracer mass from cruise altitudes to higher altitudes. To address the reviewer's comment we incorporated these lines in the revised manuscript *"Some of the upward flux from cruise altitudes in such short time could also be due to vertical advection scheme that was used in our simulations. In few sub-tropical regions and Arctic region as shown in Figure 4, we are even injecting the CAAE tracers above the tropopause region (as the tropopause is lower in arctic regions ~ 8 km) which can vertically mix the tracer to upper altitudes. As discussed in Vennam et al., 2017 some of the CAAE can get transported along the isentropes to higher altitudes when higher isentropes show an upward pattern."*

*Comment:* While understanding the experimental set-up, I do not understand how to correctly interpret the simulation results. The quantity "Mass Fraction (surface layer)" is to zero at the beginning of the simulation. During the simulation the values increase constantly (as explained by the authors). The concentration increases everywhere in the model domain without limits and the ratio between the concentrations in the surface model layer to the column will converge to the model layer air mass to the column air mass and hence independent from the emission location. Latter because the difference between concentrations of model grid points concentration is getting small compared to the absolute steadily increasing concentration. Hence it looks arbitrary to me to take out any specific point in time. I think there is a principle problem in the interpretation of the results for this simulation set-up, which can only be resolved by a change in the simulation set-up. One possibility is to define a sink at the surface (deposition, ...) in a meaningful way. The simulation will converge to a quasi-steady state. This has also the advantage of having the possibility to check whether the results are in steady- state after three months. Referring to Grewe et al 2014 Figure 9, the water vapor temporal evolution for cruise emissions show larger time scales, which might question the assumption of achieving a steady-state after 3 months (or actually 2 months, see below).

*Response:* To interpret and discuss our results we selected "Mass Fraction" metric to highlight the tracer magnitudes in various altitudes relative to the total column. Since the emissions are released near the cruise altitudes, mass fraction near the surface should be zero at the beginning of the simulation. These cruise altitude emissions get transported with time to different model grids and altitudes, which is likely to increase the concentrations in the model domain. We agree with the reviewer that the concentrations can keep increasing near the surface if we run the model for longer time periods, as we did not consider any sink process in our tracer simulations. However, it is precisely for this very reason that we ran the model for 3 months in each season to quantify the

magnitudes of the cruise altitude tracers that could get transported during the typical transport time of pollutants in the atmosphere. We acknowledge that not having sink is one of the limitations in our study but we intentionally structured it such that we are focusing mainly on the tracer transport and isolate any deposited tracer mass on the surface. The change in the simulation setup and/or change in the model will give different results as the transport processes are highly dependent on some of these factors. To address reviewer's comment we added these lines in the revised manuscript "*A key limitation in this study is that we did not consider any sink process in the tracer simulations and incorporating sink process might have given us an opportunity to run for longer time periods instead of three-month simulation. We envision future studies to address this limitation to further advance our understanding the role of CAAE on surface air quality.*"

We feel the 3-month simulation period that we considered is a reasonable time period as pollutants takes 1 – 2 days for vertical transport from PBL to the surface, ~1 week from mid-troposphere and ~1 month from tropopause (Jacob D.J, 1999, Figure 4-24). The horizontal transport in subtropics takes ~2 weeks and the transport from subtropics to tropics or towards poles takes ~1 – 2 months (Jacob D.J, 1999). Stohl et al., 2002 clearly indicated intercontinental transport occurs on timescales of weeks to 30 days and transport from lower stratosphere to lower troposphere can occur in range of ~90 days. Liang et al., 2009 demonstrated that it takes one month to cross the tropopause, one month to transport from upper troposphere to middle troposphere, and another one month to get transported to lower troposphere. Therefore, taking into consideration these relevant atmospheric transport timescales from the literature, we carried 90 days continuous tracer run in our tracer modeling to capture intercontinental, cross tropopause, and upper troposphere to lower troposphere transport processes.

Comment: Quantities are biased by the model resolution. Results are not given independently from the model resolution (see below), which inhibits a proper interpretation, even if the comment B wouldn't apply.

Response: The modeling results can vary with the considered spatial extent and temporal scales. We agree that the quantities could have been biased with the model resolution, but here we converted the concentrations in each layer to mass per area units (molecules/cm$^2$) considering the height of the each layer and compared with the total tracer column (molecules/cm$^2$) to estimate the relative fraction of tracer in different altitudes. We interpreted our results in mass-based units, which is an appropriate metric for the analysis that we showed in this paper.

Specific comments Abstract

Comment: p1 / l11 Please specify a bit more what kind of tracer you are referring to: inert gas- phase, inert particles with sedimentation, ..., what loss processes?

Response: We expanded our description of a passive tracer described in the introduction section. To enhance clarity, we explicitly included the phrase *"no chemistry and loss processes"* right next to passive tracer.

*Comment:* p1 / l12 Please explain in more detail what "tracer mass fraction means" in this con- text. Everything which is emitted in the atmosphere will eventually be deposited at the ground. From this perspective 100% would be expected. Near the surface is crucial. The smaller the volume the lower the percentage?

*Response:* To improve the clarity, we deleted the word "fraction" in the abstract (as we explained the metric clearly in the methodology section) and rephrased those lines as following "*Our results from Northern hemisphere simulations highlight that only < 0.6% of CAAE tracer mass get transported to the surface after 90 days of transport time considered in this study*" in the revised manuscript. In this context, tracer mass fraction is average percentage of CAAE tracer that can reach the surface through transport at worst-case conditions (no sink, no chemistry and continuous emissions).

We agree that everything that was emitted in the atmosphere will eventually get transported to the surface or undergo chemistry before getting deposited to the ground. With this setup, if we ran the model continuously for few months that would be the case. However we ran the model for four 90-day periods to represent each of the four seasons as a new model simulation. This gave us the ability to study the fate and transport of cruise altitude tracer in the 3-month period span during each season when chemistry and loss processes are turned off.

*Comment:* p1/l14 why "even"? It seems that the authors have expected something else. Please clarify this.

*Response:* We deleted the word "even".

*Comment:* p1/l16 Unclear. If something is emitted at 12 km it will always be deposited downwind. There is no direct instantaneous downward transport.

*Response:* Yes we agree with the reviewer that any pollutant emitted in the higher altitudes eventually gets transported, undergoes chemistry, gets deposited in the atmosphere and of course does not always undergo direct instantaneous downward transport. Since the main objective of this study is to quantify the amount of the passive tracer that get transported to the surface from cruise altitudes, so here in this line we are quantifying the source region CAAE contribution on the receptor region due to transport. To address this comment we included additional information in the revised manuscript "*The tagged source tracer simulations illustrated the source-receptor relationships and showed that ~10 – 50% source contributions can occur in downwind receptor regions due to transport when all other atmospheric processes are turned off*".

Introduction

*Comment:* p1/l20-22 There are indeed a couple of passive tracer studies, which were not included here, but a comparison might have been of interest. They are often not directly referring to the surface as a receptor region, but Figures are often including this information.

* Velthoven et al Atmospheic Environment 1997, is referred to below.

* Köhler, I., Sausen, R., Reinberger, R., Contributions of aircraft emissions to the at- mospheric NOx content, Atmospheric Environment, Volume 31, Issue 12, 1997, Pages
1801-1818,

* Danilin, et al. Aviation fuel tracer simulation: model intercomparison and implications, Geophys. Res. Lett., 25 (1998), pp. 3947-3950

* Schoeberl, M.R., Morris, G.A., 2000. A Lagrangian simulation of supersonic and subsonic aircraft exhaust emissions. Journal of Geophysical Research 105, 11,833–
11,839

* Rogers, H. L., H. Teyssedere, G. Pitari, V. Grewe, P. van Velthoven, J. Sundet, Model intercomparison of the transport of aircraft-like emissions from sub- and supersonic aircraft, Meteorol. Z., 3, 151-159, 2002.

* Grewe, V., Reithmeier, C. and D.T. Shindell, Dynamic-chemical coupling of the upper troposphere and lower stratosphere region, Chemosphere: Global Change Science, 47, 851-861, 2002.

Some of this is referred to later in the text, but should be clarified already here, since the impression is given that those studies do not exist.

*Response:* We thank the reviewer for bringing these papers to our attention. Firstly, we did not include some of this literature in the paper as few of these papers focused on the supersonic emissions which is outside the focus area of our study. But however we appreciate the reviewer's suggestion, and thus included few of them as shown below in the introduction text whenever they seem relevant. However, none of the papers cited above quantified the % of CAAE that can get transported to the surface.

*"The tagging approach (Grewe et al., 2017) and passive tracer modeling were also implemented in a few prior aircraft related studies to emphasize the role of transport on the CAAE emissions as discussed below".*

*"Few earlier as well as recent studies (Brasseur et al., 1998; Grewe et al., 2017) discussed the relative contribution of air traffic emission source to the total pollutant air quality concentrations using tagging approach and conducted brief analysis on this topic in their overall study. However they did not quantify the amount of CAAE that can get transported to the surface and the role of aircraft emissions in the intercontinental transport which is the focus of this study".*

*Comment:* p2 / l5-19 The text might suggest that aircraft emission tagging approaches were not used previously. However, there are studies 20 years back (Brasseur et al 1998) or recently Grewe et al 2017, who use such approaches. Please clarify the text.

* Brasseur et al, European scientific assessment of the atmospheric effects of aircraft emissions, Atmospheric Environment, 32, 1998, 2329-2418 (Figure 35)

- Grewe, V., Tsati, E., Mertens, M., Frömming, C., and Jöckel, P., Contribution of emis- sions to concentrations: The TAGGING 1.0 submodel based on the Modular Earth Sub- model System (MESSy 2.52), Geosci. Model Dev. 10, 2615-2633, doi:10.5194/gmd2016-298, 2017.

*Response:* We included some of the 3-D modeling related passive tracer studies in the manuscript that seem relevant to our study. We also want to clarify that our intention is not to suggest that the tagging approaches were not used in aircraft emission studies, rather to highlight that this is the one of the first studies to tag the key aviation emission source regions such as North America, Europe and East Asia and study the source-receptor contributions of these emissions. The two studies (Brasseur et al., 1998; Grewe et al., 2017) that the reviewer pointed out are also now referenced in revised manuscript as shown in an earlier response.

*Comment:* p3/l5 I suggest to use "atmospheric transport (resolved, parameterised and unre- solved)" instead of "dynamic", since the role of turbulence, diabatic heating, etc. is not investigated.

*Response:* We agree with this comment, and have incorporated the change as suggested.

*Comment:* p3 l7-9 I do not understand the importance and significance of this approach. There are several effects mixed. Continuous emission lead to continuous increase of the concentrations. At any time, the surface concentration is a snap shot and mixes emissions at time T0, where a lot of the emitted species may have reached the surface with emissions at time T0+90days, where no contribution to the surface concentration is expected. Further, since the tracer is not deposited it remains in the atmosphere and artificially increases the concentration, which disagrees with the worst case assumption.

*Response:* Yes we agree that continuous emissions can lead to continuous increase of the concentrations. In this study we consider the same continuous emissions scenario without chemistry/deposition as our worst-case scenario which provides us the opportunity to assess the influence of transport alone on cruise altitude emissions. The main objective of this worst-case scenario is to see how much of the T0 emissions get transported to the surface in a T0+90 days transport time. We believe that any passive tracer simulations are artificial modeling exercises as that wouldn't be the case in the actual atmosphere since both chemical and physical processes drive the fate of the pollutant. Since we did not consider deposition, we intentionally ran the model for only 3-month period (typical transport times as mentioned in the literature) to understand the transport and accumulation of these tracers.

Methodology

*Comment:* Section 2.1: Nothing is said about the vertical extend of the domain, number of layers and especially the resolution at tropopause levels. This is an extremely important point. Large-scale transport as well as diffusivity of the transport scheme

might give largely different results for too low vertical resolution or a too low model top. The large concentrations at 20 km might indicate such problems.

*Response:* We agree with this gap in information provided in our Methods, and have now included the vertical structure of the model in the Supplementary information in the revised manuscript. We included additional details regarding the vertical resolution in our response to the first comment in this document. We have 44 model layers with a 600 – 800 meters resolution near the cruise altitudes (9 – 12km), much finer near the boundary layer with 50 – 100 meters resolution, and between the boundary layer and the cruise altitudes the resolution is the range of 200 – 500 meters. So overall the vertical resolution is fairly fine enough to resolve the transport patterns in the atmospheric altitudes of interest in this study.

Comment: Section 2.2 p4 114 why "only"? Is there a disadvantage in this approach?

*Response:* No, it is not a disadvantage. It is only to emphasize how we structured the tagged tracer spatial extent.

*Comment*:

Equation (1): I strongly advise the authors to convert this mass fraction from an extensive to an intensive quantity. In the current version "Mass Fraction (MF-layer)" is dependent on the model resolution, which is not given (see above). For example a thick layer at 500 hPa will give a large number, not because there is a lot of tracer mass in terms of concentration, but solely because the model layer is thick. One possibility is to further divide by the layer thickness to obtain %/km as unit. The consequence is that the results are resolution independent, better to be interpreted and can be compared to other model results. In Figure 3, e.g., the total amount can be obtained by summing up the values. When MF is divided by the layer thickness, the total amount can be obtained by integration.

*Response:* We should clarify that in the Mass Fraction calculation we did consider the model layer thickness when converting the concentrations to mass per area term and note that the units of mass in each layer is molecules/cm$^2$. The Mass Fraction % is the relative mass available in each layer with respect to the total mass column (calculated by integrating the concentrations for all layers). The MF% in all layers sums up to 100% , so it is easy and convenient for the reader to understand CAAE tracer % that got transported from cruise altitudes to surface and other altitudes. Therefore we decided to retain the MF metric in the manuscript.

However, to be responsive to the reviewer's suggestion, we calculated the %MF/km metric, and included a new Figure 1 at the end of this document. Overall_since we are normalizing with the layer thickness, the values seem small. As the reviewer mentioned the total amount can be obtained by integration. The %/km metric shows higher values near the surface layer than the cruise altitude layer which doesn't look correct as the actual concentrations shows different trend (as shown in Figure 4 of the manuscript).

*Comment:* p5l11 I think, by taking the last 30 days out of a 90 days simulation period, you implic- itly assume a spin-up of 60 days and 30 days simulation, right? Hence, actually you assume a much smaller mixing time than suggested in Section 2. I suggest being a little bit more specific about the lifetimes used. For example if you assume that 95% of the air is mixed after 60 days, this will result in an e-folding mixing lifetime of 20 days, which is certainly too low. A 60 day e-folding mixing lifetime will result in a 95% mixing after 180 days, which would require a much longer simulation time.

*Response:* As we mentioned in earlier responses, the transport time that we considered are based on the literature we have cited. We quantified and presented figures for all three months for various metrics, but for few calculations (e.g. tracer mixing ratios presented in Figure 4 [and source-receptor contribution metric presented in Figure 8]), we presented only the last month's values to provide overall final numbers after giving sufficient time for transport to occur in the model.

*Comment*: Figure 1: Units are actually moles/s/gridbox. Please provide Figures, which are not dependent of the chosen resolution, e.g. moles/s/m2 or moles/s/m3.

*Response*: We realize that rather than moles it is useful to show the emission plot in mass terms (i.e., tons/year) as shown in Figure 1 of the manuscript. Our intent here is to show the overall spatial representation of the emissions rather than actual quantitative numbers so we feel tons/year/gridbox serves better for this purpose than moles/s/m2. Also, we believe that this approach to present mass/unit time without area normalization has been used in numerous ACP publications. And finally, since these emission plots are vertically summed values (column totals), moles/s/m3 is not applicable here.

*Comment:* p5 l23: How well do you simulate tropopause folds and the associated strat-trop exchange?

*Response:* We discussed the tropopause dynamics in reviewer's first comment. As mentioned before, H-CMAQ showed good representation of the pollutant vertical profile when compared with observations. Since WRF and H-CMAQ use a rigid "lid" on the top, they might not fully capture the complete stratosphere-troposphere exchange but as demonstrated by Xing et al (2016), the models are capable of simulating the tropopause folds. However, we do concur that there is a need for future studies with detailed evaluation of UTLS dynamics and tropopause folds.

*Comment:* p5 l23ff: The discussion is very speculative. I think you should be able to support your arguments with your simulation results. E.g. showing the tropopause fold etc.
*Response:* Here instead of calculating the tropopause folds we supported our findings with previous studies that showed the same trend due to tropopause folds. We corroborated our findings with another recent study (Akritidis et al., 2016) that showed the role of tropopause folds occurred at Mediterranean region during summer season.

However, performing a detailed tropopause fold calculation is beyond the current scope of this study, and might not yield anything new.

[Figure]

**Figure 1:** The overall model domain vertical profile of tracer mass fraction (%/layer thickness) at different altitudes (points on the line) in the model for each month by season.

**References:**

1) Xing, J., Mathur, R., Pleim, J., Hogrefe, C., Wang, J., Gan, C.-M., Sarwar, G., Wong, D. C., and McKeen, S.: Representing the effects of stratosphere–troposphere exchange on 3-D $O_3$ distributions in chemistry transport models using a potential vorticity-based parameterization, Atmos. Chem. Phys., 16, 10865-10877, https://doi.org/10.5194/acp-16-10865-2016, 2016.

2) Mathur, R., Xing, J., Gilliam, R., Sarwar, G., Hogrefe, C., Pleim, J., Pouliot, G., Roselle, S., Spero, T. L., Wong, D. C., and Young, J.: Extending the Community Multiscale Air Quality (CMAQ) modeling system to hemispheric scales: overview of process considerations and initial applications, Atmos. Chem. Phys., 17, 12449-12474, https://doi.org/10.5194/acp-17-12449-2017, 2017.

3) Xing, J., Mathur, R., Pleim, J., Hogrefe, C., Gan, C.-M., Wong, D. C., Wei, C., Gilliam, R., and Pouliot, G.: Observations and modeling of air quality trends over 1990–2010 across the Northern Hemisphere: China, the United States and Europe, Atmos. Chem. Phys., 15, 2723-2747, https://doi.org/10.5194/acp-15-2723-2015, 2015.

4) Daniel J. *Introduction to Atmospheric Chemistry*, Princeton University Press, 1999.

5) Stohl, A., S. Eckhardt, C. Forster, P. James, and N. Spichtinger, On the pathways and timescales of intercontinental air pollution transport, J. Geophys. Res.,

107(D23), 4684, doi:10.1029/2001JD001396, 2002.

6) Liang, Q., Douglass, A. R., Duncan, B. N., Stolarski, R. S., and Witte, J. C.: The governing processes and timescales of stratosphere-to-troposphere transport and its contribution to ozone in the Arctic troposphere, Atmos. Chem. Phys., 9, 3011-3025, https://doi.org/10.5194/acp-9-3011-2009, 2009.

7) Akritidis, D., Pozzer, A., Zanis, P., Tyrlis, E., Škerlak, B., Sprenger, M., and Lelieveld, J.: On the role of tropopause folds in summertime tropospheric ozone over the eastern Mediterranean and the Middle East, Atmos. Chem. Phys., 16, 14025-14039, https://doi.org/10.5194/acp-16-14025-2016, 2016.

8) Hogrefe, C., Liu, P., Pouliot, G., Mathur, R., Roselle, S., Flemming, J., Lin, M., and Park, R. J.: Impacts of different characterizations of large-scale background on simulated regional-scale ozone over the continental United States, Atmos. Chem. Phys., 18, 3839-3864, https://doi.org/10.5194/acp-18-3839-2018, 2018.

---

## Author Comment (AC3) · 21 Apr 2019

Comment: The paper discusses the dispersion of emissions of aviation from cruise altitudes in the Northern Hemisphere atmosphere. The paper considers the emissions as given in a data set as provided by FAA and Volpe for the ACCRI project [Wilkerson et al., 2010; Brasseur et al., 2016]. The emissions are treated as passive tracers, without any removal process in the atmosphere. The emission are followed over several seasons (3 months periods), starting from zero initial concentrations. The model considers tracer transport by advection with the resolved wind field of a hemispheric global model and by diffusion from a convective mechanism. I do not know this model and information is given in this paper about this model only in terms of a few references and some resolution information.

Response: We thank the reviewer for the thorough review and providing valuable comments that helped us to improve the manuscript. We provide below detailed responses to the review comments, and revisions to the manuscript where required.

The Community Multiscale Air Quality (CMAQ) model used here is primarily developed by the U.S. EPA, and has several thousands of users across the world from over 50 countries. According to Google Scholar, there are over 5000 publications since the year 2000 that refer to this model, and the model has gone through five different external peer reviews of its science during the development stages. As of date, there are over 25,000 downloads of the model across the world. An overview of the model, its science components and applications, and the global user community is available at: https://www.epa.gov/cmaq. A near-comprehensive list of peer-reviewed publications from 2000 – 2018 is available at: https://www.epa.gov/cmaq/cmaq-publications-and-peer-review.

To address the model and resolution information comment, we added some more detailed description about the model and the resolution used in methodology section as shown below *"The state-of-the-art EPA's Community Multi-Scale Air Quality (CMAQv4.7.1, doi:10.5281/zenodo.1079879) chemistry-transport model (Byun and Schere 2006) was used over a Northern hemispheric-wide domain at a grid resolution of $108 \times 108 \text{ km}^2$ that has spatial extent as shown in Figure 1 with 44 vertical layers of variable thickness between surface and 50 mb (Table S.1). The CMAQ model has been extensively used in numerous urban-to-regional scale air quality studies globally for both*

*research as well as regulatory applications to study the formation of several pollutants including ozone, fine particulate matter and air toxics [Foley et al.,2010; Appel et al., 2017; Astitha et al., 2017; Zhang et al., 2018]. In this study we used the new hemispheric CMAQ (Xing et al., 2015; Mathur et al., 2017) that has the capabilities to address long-range and intercontinental hemispheric pollution transport. The extended hemispheric CMAQ has also been evaluated against surface as well as aloft observation data (Mathur et al., 2017; Vennam et al, 2017; Hogrefe et al., 2018) and model processes were also examined for the new larger spatial scales. We turned off all chemistry and deposition processes in the model and turned on only the transport processes to perform the tracer simulations."*

Comment: So the paper studies how a passive tracer emitted from a more or less continuous source near the tropopause accumulates in and gets distributed over the atmosphere within 3 months periods for various seasonal meteorological conditions. The paper studies the mass fraction of tracers in model layers and in various source and receptor domains including a surface layer (of unknown vertical thickness).

Response: In the revised manuscript we included the vertical structure (as mentioned in the previous comment) used in the model along with the surface layer thickness information in the Supplementary document Table S.1 and also referenced the previous studies that provided detailed layer description in the manuscript.

Comment: The paper aims to investigate physical processes in transporting cruise altitude emissions in the atmosphere. However, I cannot learn anything about physical processes except that they vary with season and altitude, and that convection may be important in summer. That is not new.

Response: The main objective of this paper is to quantify the magnitudes of CAAE that reach surface layer due to transport and the influence of CAAE from source regions on the receptor regions. However, in a recent study (Vennam et al, JGR 2017), we were able to carefully quantify the air quality contributions ($O_3$ and $PM_{2.5}$) of CAAE at the surface both in North America and in Northern Hemisphere and show the vertical transport pathways through isentropic analysis. Since CAAE occur mainly near the tropopause, a region where isentropic mixing/transport is important and highly influenced by potential temperature, Vennam et al (2017) studied the isentropic-based aircraft-attributable concentrations for all seasons to understand the transport processes. The detailed physical processes responsible for the transport in general have been studied as part of the algorithm development in CMAQ and published elsewhere (Pleim et al., 2007a,b). However, Vennam et al did not separate the role of transport alone in how CAAE may affect surface layer concentrations.

In the current study, we were able to clearly illustrate that even at worst-case conditions (i.e., no chemistry or deposition, and with continuous CAAE at cruise altitudes) by using fine-scale horizontal grid resolution (~4 – 6 times finer than typical global models (Whitt et al., 2011)) and finer vertical resolution, only insignificant fraction of CAAE reaches the surface due to transport. Here we were also able to quantify the contribution of CAAE

from source region on the receptor regions that occurred due to intercontinental transport. Both these findings are new and were not addressed in the previous tracer studies (Kohler et al., 1997; Grewe et al., 2010). We thus believe that this paper is a valuable and unique contribution, and advancing the knowledge in understanding the role of cruise altitude aviation emissions on surface impacts.

Comment: The paper claims to be the first in using a "tagged tracer simulation" to quantify source- receptor relationships. Tagging is needed to follow the fate of tracers in a nonlinear system [Grewe et al., 2010]. In this study, the tracer transport is linear in the concentration values. A doubling of the sources causes a doubling of the concentrations. In this case, emissions from various sources can be treated independently of each other and tagging is trivial. Similar studies of the dispersion of NOx as a passive tracer from various sources, with linear chemistry, have been presented, e.g. by Ehhalt et al. [Ehhalt et al., 1992] and Köhler et al. [Köhler et al., 1997], long ago.

Response: Our intent is not to say that this is the first tagging study but to highlight that this is one of the first studies to tag cruise altitude aircraft emissions by region to study the source-receptor relationships of the CAAE tracers. To be more specific, we modified the line in the revised manuscript as following: "*this is the first study to use tagged tracer simulations for these high aircraft activity regions to illustrate the role of intercontinental transport*". The studies that the reviewer cited are different in many aspects (such as model configuration, aircraft emissions and modeling methodology) from our study. Kohler et al., 1997 considered NOx emissions with simplified linear chemistry so it is not a passive tracer study, and we already included some other additional passive tracer study references in our manuscript. Ehhalt et al., 1992 developed a very simplified 2-D approach with uniform vertical wind and their main intention was to demonstrate the importance of aircraft to NOx upper troposphere budget, which is different from what we are trying to address in this study with a 3-D model passive tracer modeling application. Note that these two studies are over 2 decades old, and since then model algorithms, emissions inventories and transport schemes are significantly advanced and improved, so it is important to address some of these research questions with evolving new modeling systems. One such example is aircraft emission inventory; previous studies used old gridded aircraft emission inventories whereas we used the new chorded highly-resolved gate-to-gate aviation emissions inventory from AEDT (Wilkerson et al., 2010) that provides better spatial and temporal representation of this source sector in the modeling system. And finally, given the growth in global aviation activity, in recent years, we find further motivation for a study like ours.

Comment: So, this is an academic study. That would be acceptable if done well. However, I also have technical problems:

Response: We did not understand the reviewer comment about academic study completely. We tried our best to address all specific comments and concerns that are explicitly pointed out by the reviewer in our responses in this document, and in the revised manuscript.

Comment: What is the vertical resolution? How thick is the surface layer? What are the time step sizes? Which process is simulated by asymmetric diffusion?

Response: In the revised manuscript, we added the layer structure in the supplementary info. Since WRF and thus CMAQ follow a sigma-coordinate system, note that the surface layer thickness varies both in space and time, and we thus provided the average surface layer thickness of ~20 meters in the manuscript. The time step size in CMAQ is considered small enough to ensure positivity and numerical stability of the solution, which satisfies the Courant- Friedrich Lewy (CFL) condition (Byun et al., 2006). Typically the time step size is 12 minutes, and the model runs the chemical and physical processes for each of this timestep and gives both an average and instantaneous concentration at the end of each hour. The asymmetric convection model (Pleim et al., 2007a,b) considered simulates the vertical diffusion in the model and Yamartino scheme (Byun et al., 2006) is chosen for the vertical and horizontal advection processes.

Comment: Page 4, line 3: why do you mention water vapor. Why not CO2?

Response: We mentioned water vapor as it is one of the highly emitted pollutant followed by NOx and CO2. In the revised manuscript to provide additional insight to the readers we included all these three pollutants. Revised line "*we considered emissions of $NO_X$ as our passive tracer since it is one of the highly emitted pollutants at cruise altitudes from aircraft, besides $CO_2$ and water vapor*". Furthermore, both water vapor and $CO_2$ are related to potential climate impacts, and given the focus of our study on surface layer concentrations (potentially related to air quality and public health), we had initially omitted $CO_2$.

Comment: More general, why do you talk about NOx emissions when you simulate the emissions as a passive tracer? NOx has a lifetime of typically 5 days in the free troposphere, and often much shorter near the surface. Thereafter, most NOx is converted to HNO3 and other species after a few days. CO2 would be closer to the passive tracer concept.

Response: The reason we considered $NO_x$ emissions is to better represent the spatial representation of the CAAE emissions that are related to surface air quality and we also mentioned this point in the revised manuscript. "*The rates of emissions of these tracers were based on actual cruise altitude $NO_X$ emissions estimates from AEDT, we considered $NO_X$ to better represent the spatial as well as temporal variation of aviation emissions in the upper layers of the atmosphere*" . Furthermore, whether it is $NO_x$ or $CO_2$, once it is treated as a passive tracer, the lifetimes are immaterial.

Comment: I am sceptical about the conservation properties of the model in this study. The paper talks about the amount of a species measured in moles. But I would expect that one should discuss a conservative concentration measure like the molar mixing ratio of the tracer (number of moles of the tracer per mole of air).

Response: The CMAQ model is formulated with vertical advection and diffusion schemes; the transport algorithms are well tested to ensure that the continuity equation is calculated to meet convergence conditions and mass is conserved (Byun et al., 2006).

We did not understand clearly the reviewer comment regarding species measured in moles. In CMAQ, the input emissions are in terms of moles for gas-phase species (or grams for particles) and converted into concentrations internally in the model. The final model outputs are in terms of mixing ratio (parts per million by volume, ppmV) and we considered these quantities to calculate mass fraction and tracer contribution in the results.

Comment: Fig. 1 presents emissions in units of moles/s. This is a species abundance source rate. In order to assess this, one needs to know the respective air volume in which the emissions occur.

Response: We may have inadvertently confused the reader here. To avoid further confusion and for completeness, we updated Fig1 (included at the end of this document) with annual emissions total plot (and changed the units to tons per year) in the revised manuscript. In CMAQ the emissions are in moles or grams (based on gas or particulate matter) and the model converts the emission units into output concentrations (mixing ratio, ppmV) considering all the necessary units conversion including the air volume.

Comment: When computing mean values, do the authors weigh the results with the volumes or do they add concentrations from small grid cells (near the poles and near the surface) with the same weight as sources from large grid cells (in the tropics and in the upper atmosphere)? That is not clear presently.

Response: We did not area-weight the concentrations and we equally weighted all the grid cells. However, since we converted the mixing ratio (concentrations) into molecules/cm$^2$ (converted into area basis instead of volume) we did consider the height of the grid in the units conversion. So that should take into account the depth of the grid in the computing mean values for each model layer and the same applies for the total vertical column calculation. Redoing the calculations with area weights is a potential refinement to this approach in the future.

Comment: Fig. 3, winter, shows a maximum of mass fraction forming in the lower troposphere, i.e., in a region without sources. How can that happen? Yes it can happen temporarily when advection dominates relative to diffusion processes. When averaged over longer period, it should not happen. However that is not discussed. I have the impression that the model violates conservation laws.

Response: The mass fraction in lower troposphere is due to the transport of the tracer from tropopause to the lower altitudes in the model. In winter, due to high westerly transport the horizontal transport in the cruise altitudes dominates and brings the tracer to low source region which can get transported to the lower altitudes simultaneously. As discussed in Vennam et al., 2017, during winter season higher isentropic surfaces get

closer to the lower troposphere isentropes, which indicates that the vertical downward transport along the isentropes are enhanced. During summer seasons, higher isentropic surfaces show an upward pattern that transports CAAE tracer to lower stratosphere.

The CMAQ model is formulated with vertical advection and diffusion schemes, the transport algorithms are well tested to ensure that the continuity equation is calculated to meet convergence conditions and mass is conserved. Both CMAQ and WRF (model used for meteorological inputs) are formulated to satisfy continuity equation and mass consistent advection (Byun et al., 2006; Pleim et al., 2007a,b).

Comment: The paper is good in citing many related studies. In fact, I was not aware on many of them. But it appears somewhat random in the selection of references (those of major and minor relevance for this paper). There are many other important studies which dealt with tracer or aviation emission transport in the global atmosphere earlier or more complete. Examples are as follows: Ehhalt et al. [1992]; Danilin et al. [1998] , Forster et al. [2003], Koehler et al. [1997], Brasseur et al. [1996], Brasseur et al. [1998], Gauss et al. [2006].

Response: We thank the reviewer for pointing out this inconsistency. In the revised manuscript we included some of these additional references, and streamlined the literature section.

Comment: In summary, the paper in its present form does not satisfy the quality criteria of ACP. I just looked at the paper Vennam et al. (JGR, 2017), cited in this paper, which just appeared. It seems that this is far more advanced. It includes the chemical processes that are mentioned in the outlook of the ACPD paper. So, I am not convinced that the present paper is still needed.

Response:  We regret that the reviewer states that this paper does not satisfy the quality criteria of ACP. While we are unsure of the specific criteria that may be referred to in this comment, we strongly believe that this is a well-founded study using a model with strong scientific credibility (with > 5000 publications to date with a global user base) on a key emissions source sector with robust conclusions that contributes to the growing body of literature on aviation air quality research, and more importantly a very relevant paper for the ACP audience. Vennam et al., 2017 (JGR) (published online 22 December 2017) studied full-flight aviation emissions (cruise altitude + landing and takeoff) impacts on the surface and the sensitivity of the grid resolution on those impacts. However that study was not able to isolate the influence of transport of cruise altitude aviation emissions (CAAE) on surface. Given that CAAE constitute a significant portion (~75% of fuel burn occurs at cruise altitudes) of total aviation emissions, and uncertainties in the cruise altitude impacts on surface, it is important to study this topic further. And in this present study we were able to specifically isolate the role of transport on CAAE emissions and their contribution at various altitudes and key source regions in the northern hemisphere using a fine scale model resolution (4 – 6 times finer than typical resolution used in most global models) and highly-resolved updated emission inventory with actual radar tracking (Olsen et al., 2013) compared to the previous studies. We thus strongly believe that this

manuscript fits the ACP criteria as it addressed some really key issues in the aviation research and advanced the current understanding of the topic studied here, focused on source – receptor relationships for cruise altitude aviation emissions.

From the ACP Subject areas, our paper focuses on the following shown in bold, again emphasizing the suitableness of this paper to the ACP audience:

| | |
|---|---|
| Subject | **Gases**, Aerosols, Clouds and Precipitation, Isotopes, Radiation, Dynamics, Biosphere Interactions, Hydrosphere Interactions |
| Research Activity | Laboratory Studies, Field Measurements, Remote Sensing, **Atmospheric Modelling** |
| Altitude Range | **Troposphere, Stratosphere**, Mesosphere |
| Science Focus | Chemistry (chemical composition and reactions), **Physics (physical properties and processes)** |

**References:**

1) Brasseur, G. P., J.-F. Müller, and C. Granier (1996), Atmospheric impact of NOx emis- sions by subsonic aircraft: A three-dimensional model study, J. Geophys. Res., 101, 1423-1428, doi: 10.1029/95JD02363.

2) Brasseur, G. P., R. A. Cox, D. Hauglustaine, I. Isaksen, J. Lelieveld, D. H. Lister, R. Sausen, U. Schumann, A. Wahner, and P. Wiesen (1998), European scientific assessment of the atmospheric effects of aircraft emissions, Atmos. Env., 32, 2329 - 2418. Brasseur, G. P., et al. (2016), Impact of aviation on climate: FAA's Aviation Climate Change Research Initiative (ACCRI) Phase II, Bull. Amer. Meteorol. Soc., 97, 561- 583, doi: 10.1175/BAMS-D-13-00089.1.

3) Danilin, M. Y., et al. (1998), Aviation Fuel Tracer Simulation: Model Intercomparison and Implications, Geophys. Res. Lett., 25, 3947 - 3950.

4) Ehhalt, D. H., F. Rohrer, and A. Wahner (1992), Sources and distribution of NOx in the upper troposphere at northern mid-latitudes, J. Geophys. Res., 97, 3725 - 3738, doi: 10.1029/91JD03081.

5) Forster, C., A. Stohl, P. James, and V. Thouret (2003), The residence times of aircraft emissions in the stratosphere using a mean emission inventory and emissions along actual flight tracks, J. Geophys. Res., 108, 8524, doi: 10.1029/2002JD002515.

6) Gauss, M., I. S. A. Isaksen, D. S. Lee, and O. A. Søvde (2006), Impact of aircraft NOx emissions on the atmosphere – tradeoffs to reduce the impact, Atmos. Chem. Phys., 6, 1529–1548.

7) Grewe, V., T. Eleni, and P. Hoor (2010), On the attribution of contributions of atmospheric trace gases to emissions in atmospheric model applications, Geosci. Model Dev., 3, 487-499, doi: 10.5194/gmd-3-487-2010.

8) Koehler, I., R. Sausen, and R. Reinberger (1997), Contributions of aircraft emissions to the atmospheric NOx content, Atmos. Env., 31, 1801-1818.

9) Wilkerson, J. T., M. Z. Jacobson, A. Malwitz, S. Balasubramanian, R. Wayson, G. Fleming, A. D. Naiman, and S. K. Lele (2010), Analysis of emission data from global commercial aviation: 2004 and 2006, Atmos. Chem. Phys., 10, 6391-6408, doi: 10.5194/acp-10-6391-2010.

10) Olsen, S. C., Wuebbles, D. J., & Owen, B. (2013). Comparison of global 3-D aviation emissions datasets. Atmospheric Chemistry and Physics, 13, 429–441.

11) Whitt, D. B., Jacobson, M. Z., Wilkerson, J. T., Naiman, A. D., & Lele, S. K. (2011). Vertical mixing of commercial aviation emissions from cruise altitude to the surface. Journal of Geophysical Research, 116(D14), D14109.

12) Pleim, J.E., 2007a: A Combined Local and Nonlocal Closure Model for the Atmospheric Boundary Layer. Part I: Model Description and Testing. *J. Appl. Meteor. Climatol.,* **46**,1383–1395, https://doi.org/10.1175/JAM2539.1

13) Pleim, J.E., 2007b: A Combined Local and Nonlocal Closure Model for the Atmospheric Boundary Layer. Part II: Application and Evaluation in a Mesoscale Meteorological Model. *J. Appl. Meteor. Climatol.,* **46**, 1396–1409, https://doi.org/10.1175/JAM2534.1

14) Byun D, Schere KL. Review of the Governing Equations, Computational Algorithms, and Other Components of the Models-3 Community Multiscale Air Quality (CMAQ) Modeling System. ASME. Appl. Mech. Rev. 2006;59(2):51-77. doi:10.1115/1.2128636.

15) Appel, K.W., Napelenok, S.L., Foley, K.M., Pye, H.O.T., Hogrefe, C., Luecken, D.J., Bash, J.O., Roselle, S.J., Pleim, J.E., Foroutan, H., Hutzell, W.T., Pouliot, G.A., Sarwar, G., Fahey, K.M., Gantt, B., Gilliam, R.C., Heath, N.K., Kang, D.W., Mathur, R., Schwede, D.B., Spero, T.L., Wong, D.C., & Young, J.O. (2017). Description and evaluation of the Community Multiscale Air Quality (CMAQ) modeling system version 5.1. *Geoscientific Model Development, 10*(4), 1703-1732. doi: 10.5194/gmd-10-1703-2017

16) Astitha, M., Luo, H.Y., Rao, S.T., Hogrefe, C., Mathur, R., & Kumar, N. (2017). Dynamic evaluation of two decades of WRF-CMAQ ozone simulations over the contiguous United States. *Atmospheric Environment*, 164. doi: 10.1016/j.atmosenv.2017.05.020

17) Foley, K., Roselle, S.J., Appel, K.W., Bhave, P., Pleim, J.E., Otte, T.L., Mathur, R., Sarwar, G., Young, J.O., Gilliam, R.C., Nolte, C.G., Kelly, J.T., Gilliland, A., & Bash,

J.O. (2010). Incremental testing of the Community Multiscale Air Quality (CMAQ) modeling system version 4.7. *Geosci. Model Dev., 3*: 205-226. doi: 10.5194/gmd-3-205-2010

18) Zhang, Y., Mathur, R., Bash, J. O., Hogrefe, C., Xing, J., and Roselle, S. J.: Long-term trends in total inorganic nitrogen and sulfur deposition in the US from 1990 to 2010, *Atmos. Chem. Phys*., 18, 9091-9106, https://doi.org/10.5194/acp-18-9091-2018 , 2018.